# *IntE*: Quantitative Framework for Qualitative Data Evaluation via Distributional Mining

## Abstract

Evaluating the quality of qualitative datasets containing responses collected for semi-structured questions is a persistent challenge. Manual analysis is slow and subjective, while existing automated methods lack a holistic, dataset-level perspective crucial for mining insights. We introduce *IntE*, a novel framework for the quantitative assessment of qualitative response datasets. *IntE* evaluates dataset quality using the cluster distributions based on collected responses and the predefined demographic distributions based on user metadata. *IntE* is structured into a four-quadrant assessment that quantifies the potential of a dataset for revealing general patterns and unique insights. The four quadrants rely on the distributions reconstructed via metadata and intra-data distances. Therefore, we propose a content-aware multi-agent system that accurately computes inter-response dissimilarity. This system features a two-stage adversarial framework for generating domain-specific evaluation instructions and an adaptive anchor algorithm to ensure scoring consistency. We validate *IntE* through controlled experiments on synthetic data, highlighting the effectiveness of its components. Additionally, a real-world social survey case study, validated by domain experts, demonstrates *IntE*'s capability to enhance knowledge discovery by accurately evaluating dataset quality and identifying key responses for analysis.

## 1 Introduction

Semi-structured questions play a crucial role in studies involving human participants. They enable researchers to gather detailed qualitative responses, uncovering *general patterns* and *unique insights* across demographic groups (e.g., age, gender). For example, in a study on student well-being, a question like "How have your eating habits changed since starting college?" might reveal general patterns, such as freshmen reporting poorer diets, as well as unique insights from unexpected cases, such as freshmen who are student-athletes maintaining healthy eating habits due to their training.

Quality evaluation of a qualitative response dataset is crucial before conducting analysis. Without it, researchers risk wasting time analyzing low-quality responses and failing to discover both general patterns and unique insights. However, existing quality evaluation methods are insufficient. Traditional manual approaches are not scalable and lack the quantitative outputs needed to guide data improvement (Mattimoe et al., 2021; Campbell et al., 2013). Most automated methods evaluate individual responses in isolation (Chen et al., 2023b; Swayamdipta et al., 2020), overlooking the inter-response relationships and overall dataset structure, which are crucial for the comparative analysis required for knowledge discovery (Fayyad et al., 1996; Glaser & Strauss, 2017).

Rather than focusing on individual responses, our work aims to evaluate a response dataset as a whole based on how much knowledge it contains for mining group-based patterns and insights. We propose *IntE*, a new quantitative quality evaluation method that assesses a qualitative dataset by comparing its internal structure to its known demographic composition. Intuitively, since the demographic structure (e.g., the age or gender distributions of respondents) of a dataset is known, the quality of the dataset can be assessed by analyzing its alignment with and divergence from that structure. Specifically, a strong alignment may suggest the dataset confirms expected, general patterns. On the other hand, a divergence between the two often signals the presence of new and unique insights. Based on this, the core idea of *IntE* is to assess a dataset by quantitatively (1) measuring the alignment and divergence between a demographic distribution based on user metadata and a cluster

distribution empirically derived from actual responses, and (2) examining the divergence within the cluster distribution itself. This dual approach allows us to identify general patterns within dense, homogeneous regions (e.g., high-density cluster centers), as well as to locate unique insights within abnormal responses (e.g., outliers and cross-demographic clusters).

More specifically, *IntE* consists of two components. First, it uses a **four-quadrant assessment framework** that operationalizes the comparison between the cluster and demographic distributions (Sec. 3.2). This framework yields four distinct metrics that characterize a dataset's utility for discovering *general patterns* versus *unique insights* at both the *distribution* and *data-point* levels. The four quadrants rely on the distributions reconstructed via metadata and intra-response distances. Therefore, *IntE* integrates a **content-aware multi-agent system** to compute nuanced, inter-response dissimilarity (Sec. 3.1). To make the agent more reliable and domain-aware, this multi-agent system enhances the LLM-as-a-Judge paradigm (Zheng et al., 2023) with two key innovations. First, a novel interactive instruction generation system enables efficient, domain-specific instruction generation (Sec. 3.1.2), moving beyond manual or black-box methods (Zhou et al., 2022; Shah, 2024; Pryzant et al., 2023; Chen et al., 2023a). Second, a dynamic anchor updating algorithm (Sec. 3.1.3) maintains a stable semantic reference frame to ensure evaluation consistency and mitigate scoring drift, improving upon static example-based approaches (Yang et al., 2024; Dong et al., 2024).

We validate *IntE* through a series of controlled experiments on synthetic data and a case study on a real-world dataset from economics research. The results demonstrate that *IntE* , the instruction generation module, and the anchor updating algorithm are effective, robust, and capable of accelerating knowledge discovery in practice. In summary, our contributions are:

- We propose a four-dimensional quantitative assessment framework for data collected from semi-structured questions, based on comparing data-driven and demographic distributions to guide researchers on data quality and potential key data.

- We design a content-aware multi-agent system to extract inter-response dissimilarity, featuring a human-in-the-loop adversarial process for instruction generation and an adaptive anchor algorithm for evaluation consistency.

- We conduct controlled experiments via a synthetic data generation system and a real-world case study, which validate the effectiveness and practical utility of *IntE* and its components.

## 2 RELATED WORK

**Quality Evaluation of Qualitative Datasets.** Traditionally, assessing the quality of qualitative datasets, such as responses from semi-structured interviews, has been a manual process centered on data richness and thickness (Mattimoe et al., 2021; Campbell et al., 2013; Ames et al., 2024; Naeem et al., 2024; Johnson et al., 2020). Since the manual process is resource-intensive and time-consuming, early computational approaches have been proposed to automate the process. However, they often failed to capture the deep semantic nuances essential for qualitative insights (Manning & Schutze, 1999; Cambria & White, 2014; Chang et al., 2009). The recent advent of LLMs has enabled significant progress, with studies showing their capacity to achieve human-level performance in assessing individual text entries against predefined criteria (Parfenova et al., 2024; Barros et al., 2025; Gilardi et al., 2023; Chen et al., 2023b). However, this data-point-level focus (Smith et al., 2014; Swayamdipta et al., 2020; Kuan & Mueller, 2022) overlooks the holistic quality of the entire dataset, which is critical for assessing its potential for knowledge discovery (Fayyad et al., 1996; Zhang et al., 2024; Reis et al., 2024; Glaser & Strauss, 2017; Ghorbani & Zou, 2019). To address this limitation, *IntE* introduces a novel framework that holistically evaluates the quality and potential of a dataset through the intrinsic cluster distribution and extrinsic demographic distribution.

**LLM-based Automated Evaluation.** Our approach is built on the "LLM-as-a-Judge" paradigm, where LLMs serve as scalable proxies for human preference judgments (Zheng et al., 2023; Kim et al., 2023; Zhu et al., 2023; Dunivin, 2024). However, this paradigm still faces several challenges. First, LLMs are susceptible to biases, including positional and verbosity biases, which can compromise evaluation objectivity (Ye et al., 2024; Li et al., 2024; Wang et al., 2023; Shi et al., 2024; Saito et al., 2023). Furthermore, ensuring consistent and reproducible judgments is difficult, as LLMs suffer from scoring drift and low self-consistency due to sensitivity to prompt phrasing (Lee et al., 2025; Zhou et al., 2024; Schroeder & Wood-Doughty, 2024). Finally, crafting high-quality,

domain-specific instructions for LLM judges also remains a labor-intensive task requiring significant expertise, which will limit rapid adaptation to new tasks (Khalid & Witmer, 2025; Raju et al., 2024). *IntE* devises a content-aware multi-agent system that enhances domain adaptability and consistency of evaluation via the interactive instruction generation and the adaptive anchor-update algorithm.

## 3 PROPOSED METHOD

Figure 1 shows the overview of *IntE* with three steps. Specifically, the input of *IntE* is a **response dataset** (Fig. 1-a) collecting free-text responses to a question, where each response is associated with user demographic metadata. Then, *IntE* uses two components to assess the quality of the response dataset, including (1) a **four-quadrant assessment framework** (the third step) that compares between the cluster and demographic distributions to quantitatively evaluate data, and (2) a **content-aware multi-agent system** (the first two steps) that enhances the LLM-as-a-Judge paradigm to compute nuanced, inter-response dissimilarity to reconstruct distributions.

In the first step, we use the **Two-stage Iterative Instruction Generation** (Fig. 1-b) system (Sec. 3.1.2) to quantify nuanced response relationships, starting with automated discovery of a universal evaluation instruction, followed by context-specific refinement via human-in-the-loop.

Second, this refined instruction is used to compute a pairwise dissimilarity matrix for the responses. To ensure consistency across these calculations, especially given the context window limitations of LLMs, we employ an **Adaptive Anchor Manifold Maintenance** (Fig. 1-c) algorithm (Sec. 3.1.3). This algorithm dynamically manages a set of reference examples, or "anchors", to provide a stable semantic frame for the LLM evaluator.

Finally, the **cluster distribution and demographic distribution are reconstructed** (Fig. 1-d). The dissimilarity matrix is used to partition the data via ensemble clustering, yielding a cluster distribution. This is compared against the demographic distribution

Figure 1: Overview for *IntE*.

derived from user metadata (Sec. 3.2.2). *IntE* formulates a **four-quadrant assessment** (Fig. 1-e) based on the delta between these two distributions, yielding metrics that characterize the dataset in terms of *general patterns* versus *unique insights* at both the *distribution* and *data-point* levels (Sec. 3.2.2).

### 3.1 CONTENT-AWARE DISSIMILARITY EXTRACTION

#### 3.1.1 PROBLEM FORMULATION

Given a response dataset $\mathcal{D} = \{d_i\}_{i=1}^N$ from a question with context $\mathcal{C}_{\text{xt}}$ (e.g., topic, evaluation axis), our goal is to extract a consistent inter-response dissimilarity matrix $\mathbf{D}$.

This requires designing a system that can: (i) generate an optimal evaluation instruction $P^*$ tailored to the context $\mathcal{C}_{\text{xt}}$, and (ii) use $P^*$ to compute dissimilarities while maintaining global consistency.

Let $\mathcal{P}$ be the space of all possible instructions. For any instruction $P \in \mathcal{P}$, an LLM-based evaluator $E$ produces a dissimilarity score $S = E(P, d_i, d_j, \mathcal{C}_{\text{xt}})$. The quality of $P$ is measured by its alignment with a reference judgment from an Oracle $O$ (a powerful LLM or human expert), providing a score $S_O$. The optimization objective is to find the instruction $P^*$ that minimizes the expected loss:

$$P^* = \arg \min_{P \in \mathcal{P}} \mathbb{E}_{(d_i, d_j) \sim \mathcal{D}} \big[ \ell(E(P, d_i, d_j, \mathcal{C}_{\text{xt}}), O(d_i, d_j, \mathcal{C}_{\text{xt}})) \big] \tag{1}$$

where $\ell(\cdot, \cdot)$ is a suitable loss or pseudo-loss function. To solve this, we model instruction generation as an optimization process using feedback as a pseudo-gradient (Sec. 3.1.2). To enforce consistency, we introduce an anchor manifold $A$, a set of reference responses that is dynamically updated (Sec. 3.1.3).

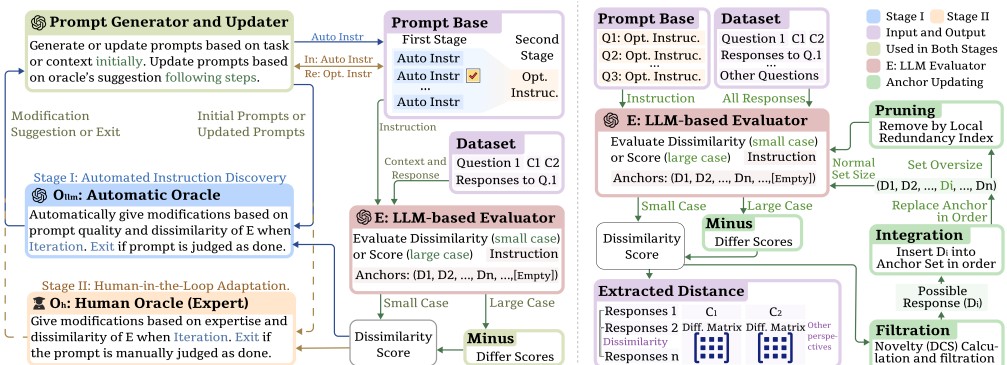

Figure 2: The Content-Aware Multi-agent System for Dissimilarity Extraction. It involves two parts: the iterative instruction generation part (Sec. 3.1.2) and the adaptive anchor manifold updating part (Sec. 3.1.3). The first part efficiently generates context-aware instructions for the second part to perform dissimilarity extraction.

### 3.1.2 ITERATIVE INSTRUCTION GENERATION

Manually authoring context-specific evaluation instructions is not scalable, as it may involve significant human labor. We propose a two-stage framework that models instruction optimization as an iterative refinement guided by feedback, which acts as a pseudo-gradient.

At each iteration $t$, an Oracle provides feedback on the current instruction $P^{(t)}$. This feedback, $\hat{\nabla}_{P^{(t)}}\mathcal{L}$, consists of a corrected score and a natural language critique, $(S_O^{(t)}, F_{\text{text}}^{(t)})$. An updater LLM, $\text{LLM}_{\text{updater}}$, uses this feedback to generate a refined instruction $P^{(t+1)}$:

$$P^{(t+1)} = \text{LLM}_{\text{updater}}(P^{(t)}, \hat{\nabla}_{P^{(t)}}\mathcal{L}) \tag{2}$$

This formulation treats the natural language critique as a pseudo-gradient, guiding the optimization in the high-dimensional space of instructions. The magnitude and direction of the "update" are determined by the content of the critique. This process unfolds in two stages:

**Stage 1: Automated Instruction Discovery.** The goal is to generate a diverse set of high-quality, general-purpose instructions. In this stage, the Oracle is a powerful LLM ($O \equiv O_{\text{LLM}}$). The iterative generation process runs automatically until a termination condition is met to build instructions from none. This condition is also determined by $O \equiv O_{\text{LLM}}$, additionally acting as a strict evaluator, which assesses the instruction against rigorous criteria for universality, clarity, and quality. The process halts when the instruction achieves a high score (e.g., greater than 0.9) and performance of LLM-based evaluator $E$ with instruction $P$. This stage concludes with a human user selecting the most promising instruction $P_a^*$ from the generated candidates.

**Stage 2: Human-in-the-Loop Adaptation.** The objective is to specialize the selected instruction $P_a^*$ for a specific task. The Oracle is a human expert ($O \equiv O_H$). $O_H$ provides precise, context-aware feedback, guiding the updater LLM to converge on a specialized instruction $P^*$. The loop terminates when the expert is satisfied with the evaluator's performance and gives no more input.

### 3.1.3 ADAPTIVE ANCHOR MANIFOLD MAINTENANCE

To ensure consistent dissimilarity measurements, we maintain a sorted anchor manifold $A = \{a_1, \dots, a_k\}$, where each anchor $a_j$ is a (response, score) pair and $k \leq k_{\max}$. It provides a stable semantic reference for the LLM. The algorithm iteratively processes new responses and updates the manifold to maximize semantic diversity while respecting the capacity constraint $k_{\max}$.

The update cycle involves three phases: **(1) Evaluation:** A new candidate response $d$ is compared against other data from $\mathcal{D}$ under the guidance of the anchor manifold to assess dissimilarity scores, which are then used to compare with the current anchors in $A$ to determine its novelty. **(2) Filtration:** Candidates that offer the most novelty are selected. Novelty is measured by the Diversity Contribution Score (DCS), defined as the average dissimilarity to existing anchors: $\text{DCS}(d) = \frac{1}{k}\sum_{j=1}^{k}\delta(d, a_j)$. **(3) Integration and Pruning:** The selected novel candidates are integrated into the manifold. If $|A| > k_{\max}$, the most redundant anchor is pruned. Redundancy is

measured by the Local Redundancy Index, $\rho(a_j) = 1 - \delta(d_j, d_{j-1}) \cdot \delta(d_j, d_{j+1})$, which identifies anchors that are semantically interpolated by their neighbors.

The implementation of the evaluation phase adapts to the dataset size $N$, but instructions for both approaches are generated through the same generation process (Sec. 3.1.2):

- **For small datasets ($N \leq N_{\text{threshold}}$):** We use a high-fidelity approach where dissimilarities $\delta(d_i, d_j)$ are computed via direct pairwise LLM calls. This has a complexity of $\mathcal{O}(N^2)$.
- **For large datasets ($N > N_{\text{threshold}}$):** To maintain computational feasibility, we use a linearized $\mathcal{O}(N)$ approach. A computationally efficient LLM operator assigns a scalar score $S(d)$ to each response. All dissimilarities are then approximated as the absolute difference between these scores: $\delta(d_i, d_j) \approx |S(d_i) - S(d_j)|$.

## 3.2 Data-Distribution-Driven Qualitative Dataset Assessment

### 3.2.1 Problem Formulation

We formalize the dataset quality assessment problem as one of comparing two different distributions of the data. Let the dataset be $\mathcal{D} = \{d_i\}_{i=1}^N$. Each data point $d_i$ is associated with:

1. An **extrinsic demographic label** $y_i \in \{1, \ldots, C\}$, derived from user metadata $\mathcal{I}_i$ via a mapping $y_i = f(\mathcal{I}_i)$. This defines the demographic partition $\mathcal{P} = \{C_j\}_{j=1}^C$., how the data distributes across different demographic partitions is called demographic distribution
2. An **intrinsic cluster label** $\hat{y}_i \in \{1, \ldots, K\}$, obtained by applying an unsupervised clustering to a dissimilarity matrix $\mathbf{D} \in \mathbb{R}^{N \times N}$. This defines the cluster partition $\hat{\mathcal{P}} = \{\hat{C}_k\}_{k=1}^K$, and how the data distributes across different cluster partitions is called cluster distribution.

**Objective:** A researcher might aim to use data either for (1) finding common responses that represent each group of people, **general patterns**, or (2) discovering previously unknown **unique insights**. These analyses can be conducted at two granularities: the **distribution level** and the **data-point level**. Aligned with these practices, we seek to define a set of normalized metrics that quantify dataset quality by evaluating the concordance and divergence between the demographic partition $\mathcal{P}$ and the cluster partition $\hat{\mathcal{P}}$, resulting in our four-quadrant assessment framework (Fig. 3).

### 3.2.2 The Four-Quadrant Assessment Framework

**(1) Defining Demographic and Cluster Distributions**

**Extrinsic Demographic Distribution:** This distribution is defined *a priori* from user metadata on characteristics researchers want to analyze. A researcher-defined function, $y_i = f(\mathcal{I}_i)$, maps attributes (e.g., expertise level) to a demographic label for each response $d_i$. The resulting vector $\mathbf{y} = \{y_i\}_{i=1}^N$ represents the ground-truth or expected structure of the dataset.

**Intrinsic Cluster Distribution:** This distribution is derived *a posteriori* from the data's internal structure. A pairwise dissimilarity matrix $\mathbf{D}$ is fed into an ensemble of clustering, where we use multiple k-means clusters and then vote for the final result (Ahmed et al., 2020). This yields a vector of cluster assignments $\hat{\mathbf{y}} = \{\hat{y}_i\}_{i=1}^N$. Since cluster labels are arbitrary, we align them with demographic labels by solving a maximum weight bipartite matching problem, where the weight between a demographic group $C_j$ and a cluster $\hat{C}_k$ is their Intersection-over-Union (Yu et al., 2016).

**(2) Preliminaries**

Let $\mathbf{y}$ and $\hat{\mathbf{y}}$ be the demographic and (aligned) cluster assignment vectors. Let $\mathbf{M} \in \mathbb{N}^{C \times C}$ be the confusion matrix where $\mathbf{M}_{jk}$ is the count of data points from demographic $j$ in cluster $k$. Let $n_j = |C_j|$ and $\hat{n}_k = |\hat{C}_k|$ be the sizes of demographic $j$ and cluster $k$. Let $\mathbf{D}$ be the dissimilarity matrix. We define:

- **Average Intra-Cluster Dissimilarity ($\bar{\delta}_{\text{intra}}$):** The mean dissimilarity between pairs of data points within the same cluster.
- **Average Inter-Cluster Dissimilarity ($\bar{\delta}_{\text{inter}}$):** The mean dissimilarity between pairs of data points in different clusters.

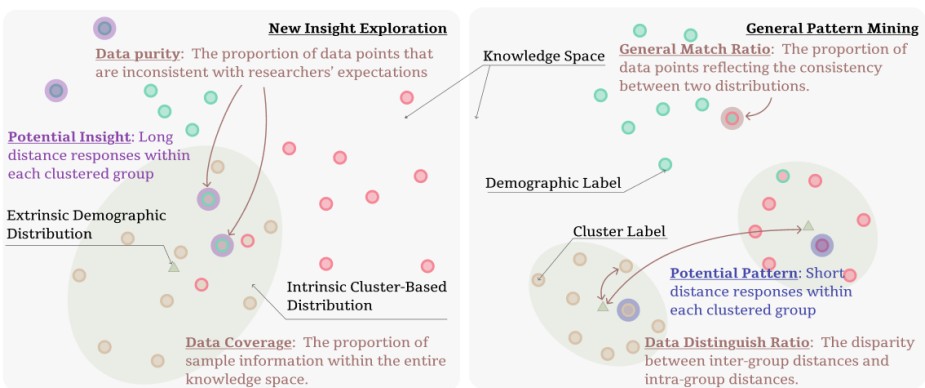

Figure 3: The Four-Quadrant Assessment Framework, which evaluates a dataset based on the research goal (General Patterns/Unique Insights) and the analysis level (Distribution/Data Point).

**(3) Evaluation Metrics**

**General Pattern / Data Point: General Match Ratio (GMR).** This metric assesses the global alignment between the demographic and cluster partitions. A high GMR indicates that the dataset's intrinsic structure aligns with its known macroscopic properties, *i.e.,* data exactly reflects behaviors of humans from different groups and is easier to find representative responses for each group.

$$\text{GMR} = \max\left(0, 1 - \frac{1}{\sqrt{C}}\left\|\alpha \cdot \frac{n_j - \mathbf{M}_{jj}}{n_j} + \frac{|n_j - \hat{n}_j|}{n_j}\right\|_2\right) \tag{3}$$

$\alpha$ is a parameter to adjust the importance of labeling errors over quantity difference. We recommend it between 1 (imbalanced dataset, emphasis unequalizing) and 2 (balanced dataset, emphasis errors).

**General Pattern / Distribution: Data Distinguishability Ratio (DDR).** It evaluates the quality of the cluster partition by measuring the separation between clusters relative to their internal compactness. High DDR shows a well-structured partitioning where clusters are coherent and distinct, *i.e.,* responses collected from different groups distinctly represent the common feature of that group.

$$\text{DDR} = \min\left(1, \beta \cdot \frac{\bar{\bar{\delta}}_{\text{inter}}}{\bar{\bar{\delta}}_{\text{intra}}}\right) \tag{4}$$

where $\beta$ is a scaling hyperparameter. We suggest setting $\beta \approx (\text{Scale} - 2\sigma)/\sigma$ with $\sigma \approx \text{Scale}/C$, reflecting the ideal ratio of maximum inter-cluster to average intra-cluster distance.

**Unique Insight / Data Point: Data Purity (DP).** It evaluates the consistency of the emergent clusters by calculating the average proportion of members that belong to the dominant demographic in each cluster. Low DP indicates that some responses containing insights that are infrequently shown in the group of interviewees like cross-group consensus or voices of underrepresented groups.

$$\text{DP} = \min\left(1, \eta \cdot \left(1 - \frac{1}{K}\sum_{k=1}^{K}\frac{|\hat{C}_k| - \max_j |\hat{C}_k \cap C_j|}{|\hat{C}_k|}\right)\right) \tag{5}$$

where $K$ is the number of clusters, and $\eta = 1$ is a scaling hyperparameter, as the core term is already a percentage-based purity metric.

**Unique Insight / Distribution: Data Coverage (DC).** This metric evaluates the conceptual dispersion of each cluster, quantifying how widely individual data points spread around the central theme of the cluster. A high DC indicates greater intra-cluster dissimilarity relative to the baseline threshold, emphasizing clusters with a broader spread, and may contain unique insights.

$$\text{DC} = \max\left(0, \frac{\bar{\bar{\delta}}_{\text{intra}}}{\gamma}\right) \tag{6}$$

where $\gamma$ is a scaling hyperparameter representing a baseline for acceptable intra-cluster dissimilarity, set relative to the typical dissimilarity $\sigma \approx \mathbf{Scale}/C$. The adjustment of $\gamma$ (e.g., $1.2\sigma$ for small datasets, $0.8\sigma$ for large) accounts for variations in dissimilarity estimation methods.

**Response Mining.** Our framework also enables the identification of specific responses for qualitative analysis. These responses, which either challenge expected structures or strongly represent general patterns, are characterized as the top-$k$ responses with the highest and lowest average dissimilarity $\bar{\delta}(d_i)$, marking them as the most central or outlier members within their clusters.

## 4 EXPERIMENT

We evaluate *IntE* through controlled experiments on synthetic data (Sec. 4.1) and a real-world case (Sec. 4.2). All experiments use "Qwen-max-latest" unless specified otherwise (Yang et al., 2025).

Controlled experiments include: 1) a user study with 20 participants to evaluate the interactive instruction generation system (Sec. 4.1.1); 2) an ablation and comparison study to validate the anchor updating algorithm (Sec. 4.1.2) and statical performance of instruction generation system; and 3) a parameter sweep study to prove *IntE* follows well under varying data quality (Sec. 4.1.3).

The case study (Sec. 4.2) applies *IntE* to qualitative responses from social science research on human behavior changing under supervision of an expert workshop, demonstrating the real-world applicability and the automatic response mining function of *IntE*.

### 4.1 CONTROLLED EXPERIMENT WITH SYNTHETIC DATA

All datasets in 4 domains (HCI, Finance, Additive Manufacturing, and Post-operative Medicine) used in these controlled experiments are produced by our Controllable Synthetic Data Generation System B.1, ensuring that ground-truth attributes are known for precise evaluation.

### 4.1.1 CONTROLLED USER STUDY FOR INTERACTIVE INSTRUCTION GENERATION SYSTEM

We conducted a controlled, within-subjects user study to assess our instruction generation system's impact on user cognitive load and efficiency.

**Experimental Setup.** We recruited 20 postgraduate students (11M/9F, aged 21-31) from diverse domains. Participants self-rated their domain experience ($M = 3.50, SD = 0.83$) and familiarity with instruction generation ($M = 3.15, SD = 1.27$) on a 5-point Likert scale. The system implements the adaptation phase of Sec. 3.1.2. The baseline condition utilized the same UI but deactivated the agent-assisted features, requiring users to compose instructions manually. Participants were tasked with generating instructions for dissimilarity extraction for two distinct questions, using both ours and the baseline. Each session lasted approximately 45 minutes and was compensated with 8 USD.

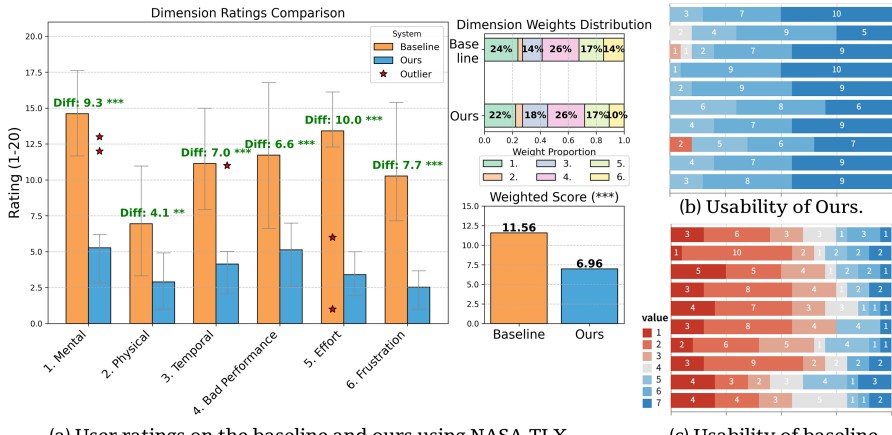

(a) User ratings on the baseline and ours using NASA-TLX.

(b) Usability of Ours.

(c) Usability of baseline.

Figure 4: Evaluation results from the user study: (a) cognitive load (higher scores = higher load); (b)–(c) system usability (7 = good usability).

We evaluated both systems on *Cognitive Load* (NASA-TLX) (Hart & Staveland, 1988) and *System Usability*. Usability was measured with a custom 10-item questionnaire on a 7-point Likert scale, assessing: Initial-Prompt Quality (Q1), Effortless-Tuning (Q2), Focus-on-Intent (Q3), No-Prompt-Expertise-Needed (Q4), Rapid-Iteration (Q5), Self-Confidence (Q6), High-Satisfaction (Q7), Creative-Insights (Q8), Easy-Learning (Q9), and Will-to-Reuse (Q10).

**Results and Analysis.** As shown in Fig. 4, our framework significantly reduces user cognitive load when composing instructions. It also demonstrates superior performance across all evaluated usability dimensions. Instructions collected from this study were used in subsequent experiments.

### 4.1.2 ABLATION AND COMPARISON STUDY

We performed an ablation study to validate the constituent components of our adaptive anchor updating algorithm (Sec. 3.1.3) and conducted comparison experiments to assess the effectiveness of the interactive instruction generation framework (Sec. 3.1.2).

**Experimental Setting.** On four 15-interviewee dataset from different domains(generated via Sec. B.1), we evaluated several ablations of our full algorithm (`Full`): disabling in-context learning (`No Example`), disabling the algorithm (`Fixed Example`), including all responses as candidates (`No Novelty Check`), removing example length limits (`No Pruning`), and using a heuristic for score calculation (`Heuristic Quality`), *i.e.,* score is the average of samples with the smallest difference.

The framework was also tested with instructions generated under three conditions, including small scale on 15, large scale on 15 and large scale on 90-interviewee datasets (generated via Sec. B.1). Experiments included several LLMs: Qwen-max-latest, Qwen-plus-latest, Qwen-turbo-latest, GLM-4.5, and deepseek-v3 (Yang et al., 2025; Zeng et al., 2025; Liu et al., 2024). We measured Pearson and Spearman correlation between extracted and ground-truth dissimilarity scores. Due to the extensive number of experiments, each reported value is the average performance across four randomly selected questions from the four domains.

Table 1: Result of the ablation study and comparison study. **Bold and underlined** represents top 1 result and **Bold** represents top 2 result in ablation study. "Direct" is the manual version and "scale up" is the result performed on the 90-dataset.

| | Direct | | Baseline | | No example | | Fix Example | | No Novelty Check | | No Pruning | | Heuristic Quality | |
|---|---|---|---|---|---|---|---|---|---|---|---|---|---|---|
| | | | | | | | Ablation Study on Small Scale (Comparision) Version | | | | | | | |
| | Spear. | Pears. | Spear. | Pears. | Spear. | Pears. | Spear. | Pears. | Spear. | Pears. | Spear. | Pears. | Spear. | Pears. |
| Qwen_turbo | -0.083 | -0.035 | **0.708** | **0.681** | 0.644 | 0.656 | 0.659 | 0.663 | 0.604 | 0.625 | 0.620 | 0.604 | 0.606 | 0.640 |
| Qwen_plus | 0.228 | 0.292 | **0.725** | **0.718** | 0.674 | 0.659 | **0.760** | **0.757** | 0.700 | 0.683 | 0.652 | 0.667 | 0.643 | 0.636 |
| Qwen_max | -0.036 | -0.013 | **0.725** | **0.728** | 0.633 | 0.621 | 0.672 | 0.663 | 0.683 | 0.695 | 0.691 | 0.697 | 0.701 | 0.709 |
| GLM | 0.599 | 0.621 | **0.830** | **0.836** | 0.770 | 0.765 | 0.822 | 0.829 | 0.826 | 0.839 | **0.831** | **0.842** | 0.729 | 0.732 |
| deepseek-v3 | 0.331 | 0.356 | **0.726** | 0.705 | 0.674 | 0.685 | 0.717 | **0.722** | 0.697 | 0.673 | 0.703 | 0.676 | **0.739** | **0.734** |

| | Scale up on Large Dataset | | Baseline | | No example | | Fix Example | | No Novelty Check | | No Pruning | | Heuristic Quality | |
|---|---|---|---|---|---|---|---|---|---|---|---|---|---|---|
| | | | | | | | Ablation Study on Large Scale (Direct Assigning) Version | | | | | | | |
| | Spear. | Pears. | Spear. | Pears. | Spear. | Pears. | Spear. | Pears. | Spear. | Pears. | Spear. | Pears. | Spear. | Pears. |
| Qwen_turbo | 0.789 | 0.805 | 0.747 | **0.775** | 0.705 | 0.733 | 0.692 | 0.721 | 0.699 | 0.730 | 0.729 | 0.765 | 0.709 | 0.732 |
| Qwen_plus | 0.813 | 0.824 | 0.719 | 0.741 | **0.796** | **0.810** | 0.735 | 0.756 | 0.757 | 0.777 | 0.762 | 0.780 | **0.767** | **0.790** |
| Qwen_max | 0.786 | 0.808 | **0.789** | **0.816** | 0.773 | 0.797 | 0.768 | 0.794 | 0.767 | 0.791 | 0.776 | 0.803 | 0.752 | 0.779 |
| GLM | 0.844 | 0.865 | **0.797** | **0.819** | 0.771 | 0.785 | 0.758 | 0.786 | 0.792 | 0.817 | 0.780 | 0.803 | 0.794 | 0.818 |
| deepseek-v3 | 0.824 | 0.840 | **0.827** | **0.852** | 0.794 | 0.822 | **0.856** | **0.878** | 0.822 | 0.846 | 0.826 | 0.850 | 0.811 | 0.839 |

**Results and Analysis.** As shown in Tab. 4.1.2, the full algorithm and its components are effective. The instruction generation system provides a statistically significant improvement in instruction quality over manual generation. The anchor updating algorithm maintains high performance as dataset size increases. While the large-scale version yielded superior results, we posit that the small-scale version may better model according to the following discussion.

**Discussion on Large-Scale Approximation:** 1. While large-scale relevance is enhanced, scalar projection simplifies high-dimensional semantic relationships into a single value, causing information loss. But this trade-off is essential for achieving efficiency. 2. This approach is applicable only when responses can be reduced to a score, such as cases influenced by personal literacy levels. Otherwise, the small-scale version can only be chosen.

### 4.1.3 PARAMETER SWEEP EXPERIMENT FOR *IntE* EVALUATION

We evaluated the overall performance of *IntE* by applying it to synthetic datasets with controlled distributional variations to validate that its behavior aligns with our design principles.

**Experimental Setting.** We generated two sets of 15-interviewee datasets (per Sec. B.1). In the first set (`Fixed Mean`), the inter-community mean difference was fixed while variance was swept from 1 to 40. In the second set (`Fixed Variance`), variance was fixed at 20 while the inter-community mean difference was swept from 5 down to 45. We applied *IntE* in both small-scale and large-scale configurations to observe how its four evaluation scores responded.

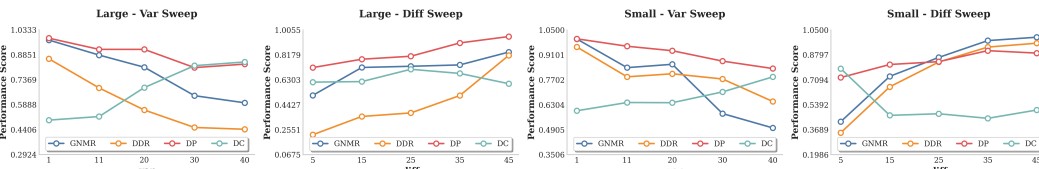

Figure 5: Result of parameter sweep experiment.

**Results and Analysis.** Shown in Fig. 5, with increasing variance, the GNMR, DP, and DDR scores decreased, while the DC score increased. Conversely, as the mean difference decreased, GNMR, DP, and DDR scores increased, while the DC score remained stable. These trends are consistent with our design, validating the behavioral correctness of *IntE*'s scoring functions.

## 4.2 CASE STUDY ON STUDENT FOOD CHOICES

To demonstrate real-world applicability, we conducted a case study using a dataset of 126 qualitative responses on college students' food preferences [1]. The task, involving three domain experts, was to relate student grade level (37 freshmen, 32 sophomores, 26 juniors, 27 seniors) to self-reported changes in eating habits since starting university. The experts used our instruction generation system (Sec. 3.1.2) to create evaluation instructions for *IntE*, aiming to assess data quality and automatically identify key responses with general patterns or unique insights.

Following the recommended parameters in Sec. 3.2.2 ($\alpha = 1.0, \beta = 1.5, \eta = 25, \delta = 1.0$), the resulting metrics ($GMR = 0.39, DDR = 0.03, DC = 0.90, DP = 0.43$) faithfully represented the data's quality. The analysis revealed a similar distribution of responses across all grade levels, "getting worse" ($mean = 61.69\%, std = 3.87\%$), "keeping the same" ($mean = 6.33\%, std = 3.61\%$), and "getting better" ($mean = 29.74\%, std = 4.89\%$). This uniformity between groups explains the low $DDR$ and $GMR$ scores. Meanwhile, a high mean across demographic groups and uncleaned raw data led to high $DC$ and low $DP$ values.

The experts reviewed the responses given by the mining function in our system, including the top-3 most general and top-3 most unique points. They confirmed the effectiveness of *IntE* in surfacing these data.

## 5 CONCLUSION

In this paper, we introduced *IntE*, a novel framework to quantitatively evaluate qualitative datasets by measuring the divergence and alignment between an extrinsic demographic distribution and an intrinsic cluster distribution. *IntE* holistically assesses a dataset's potential for yielding both general patterns and unique insights, enabled by a four-quadrant assessment framework and a content-aware multi-agent system that computes robust dissimilarity scores using interactive instruction generation and adaptive anchors. Our empirical evaluation, through controlled experiments and a real-world case study, validated *IntE*. The results demonstrate that *IntE* effectively assesses dataset quality and accelerates knowledge discovery by automatically surfacing high-value responses for analysis.

---

[1] https://www.kaggle.com/datasets/borapajo/food-choices

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

## A USE OF LLM

In this paper, we used Gemini to check grammar and improve wording. It did not change the original meaning of the text or introduce any new references or knowledge.

We also used it to search for related work. All retrieved papers were read and reviewed by the authors, who manually decided whether to include them.

## B CONTROLLABLE SYNTHETIC DATA GENERATION

We use the following controllable synthetic data generation system B.1 to controllably generate data for experiments, ensuring that ground-truth attributes are known for precise evaluation. In the experiment, we generated 4 datasets from different domains, including finance, medical, human-computer interaction, and crowd-sourcing for supervised finetuning data collection. The questions are listed below.

## B.1 CONTROLLABLE SYNTHETIC DATA GENERATION SYSTEM

We generate synthetic data with precise ground-truth attributes using a modular, four-stage pipeline of LLM-based agents.

**Community Definition.** Given a questionnaire $Q$, an Architect Agent defines $k$ evaluation dimensions $D$ and $m$ user communities $C$ (e.g., "expert", "novice"). For each community $c_j$, it generates a qualitative profile $P_j$ describing its background and behaviors.

**Quantitative Score Generation.** For each synthetic user $i$ in a community $c_j$, we generate a ground-truth score vector $s_i \in \mathbb{R}^k$. Each score $s_{i,k}$ for dimension $d_k$ is sampled from a specified distribution (e.g., $\mathcal{N}(\mu_{j,k}, \sigma_{j,k}^2)$), where the parameters $(\mu_{j,k}, \sigma_{j,k})$ are defined per community to control attribute levels and diversity.

**Persona Instantiation.** A Persona Agent creates a narrative persona $p_i$ conditioned on the community profile $P_j$ and score vector $s_i$. The persona's background (e.g., education, career) is generated to be consistent with the assigned scores: $p_i = \text{AgentPersona}(P_j, s_i)$.

**Response Simulation.** An Interviewee Agent simulates the user's responses $R_i$ to the questionnaire $Q$. The generation is conditioned on the persona $p_i$ and score vector $s_i$, ensuring the responses reflect the ground-truth scores. The score vector $s_i$ is prioritized to ensure quantitative control: $R_i = \text{AgentInterviewee}(p_i, s_i, Q)$.

For our experiments (Sec. 4.1), we applied this system to generate response sets for four questionnaires (HCI, Finance, Additive Manufacturing, and Post-operative Medicine) with quantitatively controlled attribute distributions.

## B.2 QUESTIONNAIRE

We provide the Sim-structured questionnaire in different domains here, which are used for dataset generation in the controlled experiment.

### B.2.1 FINANCE QUESTIONNAIRE

**Questionnaire Name:** Finance

**Questions:**

1. **Questions related to banking services**
   - **Financial behavior:** How often do you engage in banking services? What types of services do you typically use at the bank?
   - **Financial literacy:** What risks do you think are associated with keeping money in the bank?

2. **Concerns about financial status**
   - **Financial behavior:** Are you concerned about your financial status (e.g., frequently checking your bank account and income/expenditure status)?
   - **Financial attitude:** How important do you think it is to monitor your financial status and why?

3. **Questions related to financial education**
   - **Financial literacy:** What activities in your daily life do you think are related to finance?
   - **Financial attitude:** Do you think financial education in schools is important?

4. **Questions about saving habits**
   - **Financial behavior:** Do you have good saving habits?
   - **Financial literacy:** Do you know what methods or channels are available for saving money?
   - **Financial attitude:** How important do you think saving money is?

5. **Perception of investment risks**

- **Financial literacy:** What do you think about the statement "high-yield investments come with high risks?" Generally, how does the risk of investment change when investors diversify their investments across different categories?
- **Financial behavior:** Do you pay attention to major financial news, such as stock market fluctuations? If you participate in investments, would you take actions to diversify your investments across different categories?

### B.2.2 MEDICAL QUESTIONNAIRE

**Questionnaire Name:** Analysis of Health Post-Treatment with New Bone Medication

**Questions:**

1. **Questions about new or unexpected health changes**

   - **Symptom discovery:** Thinking back to the time after your bone started feeling better, what was the very first new or unexpected health change you noticed that was not related to your original injury? Please describe what it was and what it felt like.
   - **Symptom inventory:** Besides that first change, what other new health concerns have you experienced since taking the new medication? Please list them and briefly describe each one (e.g., skin rashes, constant tiredness, digestive issues, headaches, mood changes).

2. **Questions about the timing and patterns of your new condition**

   - **Timing and onset:** For the main new health issue you mentioned, can you pinpoint when it started in relation to when you were taking the new medicine? (e.g., "It started a week after I began the medication," or "It didn't show up until a month after I finished the full course").
   - **Pattern and triggers:** Have you noticed any patterns to this new condition? For example, is it constant, or does it come and go? Is there anything that seems to make it better or worse (like certain foods, activities, stress, or time of day)?

3. **Questions about the impact of the new condition on your life**

   - **Impact comparison:** Think about the challenges of the original broken bone versus the challenges of this new health condition. Which one has had a bigger impact on your day-to-day life, and why?
   - **Life adjustments:** What is one specific activity or part of your daily routine that you've had to change or give up, not because of your bone, but because of this new health condition?

4. **Questions about your own thoughts and actions regarding the new condition**

   - **Personal hypothesis:** Before this questionnaire, had you already made a connection in your own mind between the new medicine and your new health condition? What made you suspect (or not suspect) a link?
   - **Communication with doctors:** Have you discussed this new health issue with a doctor before? If so, what was that conversation like? If not, what has held you back from bringing it up?

5. **Questions about your overall perspective on the treatment**

   - **Future decision-making:** How has this experience changed how you will approach taking a new or experimental medication in the future? What questions would you ask your doctor now that you might not have asked before?
   - **Defining successful treatment:** This medicine successfully healed your bone, but may have caused other issues. How does this experience change your personal definition of a "successful" medical treatment?

### B.2.3 HUMAN-COMPUTER INTERACTION QUESTIONNAIRE

**Questionnaire Name:** Human-Computer Interaction (HCI)

**Questions:**

1. **AI System Interaction Experience Questions**

   - **AI Response Expectations:** Does the AI system's response meet your expectations?
   - **AI Understanding Accuracy:** Do you feel that the AI understands your needs accurately when using the AI system?
   - **Desired AI Features:** What features would you like the AI system to add to enhance user experience?

2. **Input Device Usage Experience Questions**

   - **User Preferences:** Does the current input device meet your operating habits?
   - **Usability Challenges:** Have you encountered any inconveniences while using the input device?
   - **Design Improvements:** What improvements would you like to see in the design of the input device?

3. **Haptic Feedback Technology Experience Questions**

   - **Haptic Impact on Experience:** Does haptic feedback have a significant impact on your experience in virtual reality?
   - **Realism of Haptics:** Do you feel that the current haptic feedback technology is realistic enough?
   - **Haptic Technology Improvements:** In what areas would you like to see improvements in haptic feedback technology?

4. **Adaptive Interface Functionality Questions**

   - **Adaptive Interface Responsiveness:** Does the adaptive interface of the system effectively respond to your changing needs?
   - **Adaptive Interface Issues:** Have you encountered any issues during the adaptive process?
   - **Desired Features in Adaptive Interfaces:** What features would you like to see added to enhance the adaptability of the interface?

5. **Multimodal Interaction Experience Questions**

   - **Efficiency of Multimodal Interaction:** Has multimodal interaction improved your interaction efficiency with the system?
   - **Multimodal Integration Issues:** Have you encountered any issues with poor integration while using multimodal interaction?
   - **Multimodal Interaction Improvements:** In what areas would you like to see further improvements in the multimodal interaction experience?

### B.2.4 CROWDSOURCING FOR SFT COLLECTION QUESTIONNAIRE

**Questionnaire Name:** 3D Printing and Additive Manufacturing

**Questions:**

1. **Questions about what 3D printing is**

   - **Simple Explanation:** If you had to explain 3D printing to a child, what would you say? How is it different from just printing a picture on a piece of paper?
   - **Core Advantage:** Think about making something by starting with a block of material and carving parts away, versus 3D printing, which builds something up from nothing. What do you feel is the biggest advantage of building things up layer by layer?

2. **Questions about the 'stuff' used in 3D printers**

   - **Everyday Materials:** If you could 3D print an object for your kitchen, like a custom spoon or a container, what qualities would the material need to have? (e.g., should it be flexible, strong, heat-resistant, etc.). Describe your ideal material in simple terms.
   - **User-Friendly Ideas:** 3D printers can sometimes be tricky to use. If you were asked to design a 3D printer for a complete beginner, what is one feature you would add to make it super easy and fun to use, even if you make a mistake?

3. **Questions about how 3D printing is used in the real world**

   - **Community Problem Solving:** Imagine your local community was given a powerful 3D printer. What is one local problem (related to parks, schools, or helping neighbors) that you think could be solved by printing something new?

   - **Factory vs Home:** Do you think we will ever 3D print everything we need at home, or will we always need big factories? Explain your thoughts on what factories will always be better at making.

4. **Questions about designing things for 3D printing**

   - **Future of Design:** 3D printing can create very complex, web-like, or hollow shapes that are both lightweight and strong. How might this change the look and feel of everyday items, like furniture, shoes, or bicycles, in the future?

   - **Smart Design Analogy:** Some software can cleverly redesign a solid part, removing all the inside material that isn't needed for strength, kind of like how nature designs a tree or a bone. What everyday example would you use to explain this idea of making things "smartly hollow" to someone?

5. **Questions about the future of 3D printing for society**

   - **Optimist or Skeptic:** Thinking about the future, are you more excited or more worried about everyone having access to 3D printers? Briefly explain what makes you feel that way.

   - **Rules and Safety:** If anyone can print anything, what is one important rule you think society or governments should consider? Think about safety, fairness, or new kinds of problems that could arise.

## C    DETAILED ALGORITHMS FOR *IntE* FRAMEWORK

The following tables provide a corrected and detailed workflow of the *IntE* framework. This includes the overall framework (Algorithm 1), the iterative instruction generation process (Algorithm 2), and the logically revised adaptive anchor manifold maintenance process (Algorithm 3), which now correctly integrates computation and maintenance.

### C.1    OVERALL *IntE* FRAMEWORK

This algorithm details the complete process for *IntE*, from instruction generation to metric calculation and response mining, with corrected metric names and descriptions based on the source paper.

---

**Algorithm 1** *IntE*: Data-Distribution-Driven Assessment Framework (Corrected)

---

1: **Input:** Response Dataset $D$, Demographic Metadata $I$, Mapping Function $f$.
2: **Output:** Metrics (GMR, DDR, DP, DC) and Key Responses (Patterns, Insights).

    **Phase 1: Dissimilarity Extraction**
3: Generate an optimal instruction $P^*$ using **Iterative Instruction Generation** (Algorithm 2).
4: Compute the dissimilarity matrix $\mathcal{D}$ using $P^*$ and **Adaptive Anchor Manifold Maintenance** (Algorithm 3).

    **Phase 2: Distribution Comparison**
5: Define demographic distribution $\mathcal{P}$: Compute labels $y_i = f(I_i)$.
6: Define cluster distribution $\hat{\mathcal{P}}$: Cluster $\mathcal{D}$ to obtain cluster labels $\hat{y}$.
7: Align $\hat{y}$ to $y$ using bipartite matching on Intersection over Union (IoU).

    **Phase 3: Assessment and Mining**
8: Compute the confusion matrix $M$ from $y$ and aligned $\hat{y}$.
9: Calculate the following metrics:
10:     **GMR (General Match Ratio):** Measures global alignment between $\mathcal{P}$ and $\hat{\mathcal{P}}$.
11:     **DDR (Data Distinguishability Ratio):** Evaluates inter-cluster separation versus intra-cluster compactness.
12:     **DP (Data Purity):** Assesses homogeneity within emergent clusters.
13:     **DC (Data Coverage):** Measures the conceptual dispersion within clusters; high DC indicates high diversity.
14: Perform response mining:
15:     **Insights:** Identify outliers (responses with the highest average dissimilarity $\bar{\delta}(d_i)$).
16:     **Patterns:** Detect archetypes (responses with the lowest average dissimilarity $\bar{\delta}(d_i)$).
17: **return** Metrics and Key Responses.

---

## C.2 ITERATIVE INSTRUCTION GENERATION

This algorithm details the process for refining instructions, clarifying the nature of the feedback provided by the Oracle.

---

**Algorithm 2** Iterative Instruction Generation

---

1: **Input:** Instruction seed, Dataset $D$, Context $Cxt$.
2: **Output:** Optimized instruction $P^*$.

    **Stage 1: Automated Discovery**
3: **Initialize:** $None$ as initial prompt; use Oracle $O_{LLM}$, Updater $LLM_{updater}$, and Evaluator (also $O_{LLM}$).
4: **repeat**
5:     Sample response pairs $(d_i, d_j) \sim D$.
6:     $O_{LLM}$ provides feedback $\nabla \hat{P}^{(t)} L$ (e.g., a corrected score and natural language critique).
7:     Update $P^{(t+1)} \leftarrow LLM_{updater}(P^{(t)}, \nabla \hat{P}^{(t)} L)$.
8:     Oracle $O_{LLM}$ scores the updated instruction $P^{(t+1)}$.
9: **until** Evaluator score reaches a high threshold (e.g., ¿0.9).
10: Human user selects the best instruction $P_a^*$ from the automated results.

    **Stage 2: Human-in-the-Loop Adaptation**
11: **Initialize:** $P^{(0)} \leftarrow P_a^*$; use Oracle $O_H$ (human expert).
12: **repeat**
13:     Present results generated by $P^{(t)}$ to $O_H$.
14:     $O_H$ provides feedback $\nabla \hat{P}^{(t)} L$. Stop if no feedback is given.
15:     Update $P^{(t+1)} \leftarrow LLM_{updater}(P^{(t)}, \nabla \hat{P}^{(t)} L)$.
16: **until** Human expert is satisfied with the instruction's performance.
17: **return** Optimized Instruction $P^* \leftarrow P^{(t+1)}$.

---

## C.3 ADAPTIVE ANCHOR MANIFOLD MAINTENANCE

This algorithm is substantially revised to correctly reflect the paper's logic, where dissimilarity computation and anchor maintenance are an integrated, iterative process. The anchor manifold is used as a semantic reference during evaluation, and key steps like sorting have been added.

---

**Algorithm 3** Adaptive Anchor Manifold Maintenance (Corrected)

---

1: **Input:** Dataset $D = \{d_1, ..., d_N\}$, Optimized Instruction $P^*$, Maximum anchors $k_{max}$, Threshold size $N_{threshold}$.
2: **Output:** Dissimilarity matrix $\mathcal{D}$.

**Initialization**
3: Initialize anchor manifold $\mathcal{A} = \emptyset$.
4: Initialize dissimilarity matrix $\mathcal{D}$ as an $N \times N$ zero matrix.
5: **if** $N > N_{threshold}$ **then**
6:      Initialize an array $S$ of size $N$ to store scalar scores.
7: **end if**

**Unified Computation and Maintenance Cycle**
8: **for** each response $d_i \in D$ **do**
9:      % – Core Computation Step –
10:      **if** $N \leq N_{threshold}$ (Small Dataset) **then**
11:          **for** each response $d_j$ where $j > i$ **do**
12:              % LLM call uses anchor manifold $\mathcal{A}$ as context for consistency
13:              Compute $\delta(d_i, d_j) = \text{LLM}(P^*, d_i, d_j, \mathcal{A})$.
14:              $\mathcal{D}_{ij} = \mathcal{D}_{ji} = \delta(d_i, d_j)$.
15:          **end for**
16:      **else** (Large Dataset)
17:          % LLM call also uses $\mathcal{A}$ as context to ensure consistent scoring
18:          Compute scalar score $S(d_i) = \text{LLM}(P^*, d_i, \mathcal{A})$.
19:      **end if**
20:      % – Anchor Manifold Update Step –
21:      Compute Diversity Contribution Score $DCS(d_i) = \frac{1}{|\mathcal{A}|} \sum_{a_k \in \mathcal{A}} \delta(d_i, a_k)$.
22:      Find anchor $a_{min}$ in $\mathcal{A}$ with the lowest DCS.
23:      **if** $DCS(d_i) > DCS(a_{min})$ **then**
24:          **if** $|\mathcal{A}| < k_{max}$ **then**
25:              Add $d_i$ to the anchor manifold: $\mathcal{A} \leftarrow \mathcal{A} \cup \{d_i\}$.
26:              **Sort** $\mathcal{A}$ (e.g., by score $S(a_k)$ or avg. dissimilarity).
27:          **else**
28:              Add $d_i$ to the anchor manifold: $\mathcal{A} \leftarrow \mathcal{A} \cup \{d_i\}$.
29:              **Sort** $\mathcal{A}$ (e.g., by score $S(a_k)$ or avg. dissimilarity).
30:              Compute Redundancy Index $\rho(a_j) = 1 - \delta(a_j, a_{j-1}) \cdot \delta(a_j, a_{j+1})$ for each $a_j \in \mathcal{A}$.
31:              Remove the anchor $a_{redundant}$ with the highest $\rho(a_j)$ from $\mathcal{A}$.
32:          **end if**
33:      **end if**
34: **end for**

**Finalize Dissimilarity Matrix**
35: **if** $N > N_{threshold}$ (Large Dataset) **then**
36:      **for** $i = 1$ to $N$, $j = 1$ to $N$ **do**
37:          $\mathcal{D}_{ij} = |S(d_i) - S(d_j)|$.
38:      **end for**
39: **end if**
40: **return** Dissimilarity matrix $\mathcal{D}$.

---

# D USER STUDY

As stated above, we conducted a user study to prove the efficiency of our instruction generation system. In this section, we will state the detailed information for it, including the user study schedule especially how to do the within-subject experiment 2, statistical significance results 3, user interface D.3, participant demographics 4, and the questionnaire we used in our experiment D.5.

## D.1 WITHIN SUBJECT USER STUDY SCHEDULE

The following information is the schedule of our study, which states how we assign people to each group and how we arrange the people to meet the balance requirement.

Table 2: Overview of User Study Schedule

| ID | Group | Participant ID | Method Order | Task 1 | Task 2 | Time |
|----|-------|----------------|--------------|--------|--------|------|
| 1 | Finance | Participant 1.1 | A → B | Q1 (Method A) | Q2 (Method B) | 1h |
| 2 | Finance | Participant 1.2 | A → B | Q2 (Method A) | Q3 (Method B) | 1h |
| 3 | Finance | Participant 1.3 | A → B | Q3 (Method A) | Q1 (Method B) | 1h |
| 4 | Finance | Participant 1.4 | B → A | Q4 (Method B) | Q5 (Method A) | 1h |
| 5 | Finance | Participant 1.5 | B → A | Q5 (Method B) | Q4 (Method A) | 1h |
| 6 | HCI | Participant 2.1 | A → B | Q6 (Method A) | Q7 (Method B) | 1h |
| 7 | HCI | Participant 2.2 | A → B | Q7 (Method A) | Q8 (Method B) | 1h |
| 8 | HCI | Participant 2.3 | A → B | Q8 (Method A) | Q9 (Method B) | 1h |
| 9 | HCI | Participant 2.4 | B → A | Q6 (Method B) | Q10 (Method A) | 1h |
| 10 | HCI | Participant 2.5 | B → A | Q10 (Method B) | Q9 (Method A) | 1h |
| 11 | Crowdsourcing | Participant 3.1 | A → B | Q11 (Method A) | Q12 (Method B) | 1h |
| 12 | Crowdsourcing | Participant 3.2 | A → B | Q12 (Method A) | Q11 (Method B) | 1h |
| 13 | Crowdsourcing | Participant 3.3 | B → A | Q13 (Method B) | Q14 (Method A) | 1h |
| 14 | Crowdsourcing | Participant 3.4 | B → A | Q14 (Method B) | Q15 (Method A) | 1h |
| 15 | Crowdsourcing | Participant 3.5 | B → A | Q15 (Method B) | Q13 (Method A) | 1h |
| 16 | Medical | Participant 4.1 | A → B | Q18 (Method A) | Q17 (Method B) | 1h |
| 17 | Medical | Participant 4.2 | A → B | Q17 (Method A) | Q18 (Method B) | 1h |
| 18 | Medical | Participant 4.3 | B → A | Q16 (Method B) | Q19 (Method A) | 1h |
| 19 | Medical | Participant 4.4 | B → A | Q19 (Method B) | Q20 (Method A) | 1h |
| 20 | Medical | Participant 4.5 | B → A | Q20 (Method B) | Q16 (Method A) | 1h |

## D.2 STATISTICAL SIGNIFICANCE OF USER STUDY RESULTS

The following table (Tab. 3) shows how much and to what extent our system overcomes the baseline system in different aspects.

Table 3: Statistical Comparison Between Baseline and Lancet

| Dimension | Baseline Mean | Ours Mean | Mean Difference | t-statistic | p-value | Significant |
|-----------|---------------|-----------|-----------------|-------------|---------|-------------|
| 1 | 3.30 | 6.35 | 3.05 | 6.45 | 0.00000 | Yes |
| 2 | 3.35 | 5.85 | 2.50 | 6.69 | 0.00000 | Yes |
| 3 | 3.00 | 6.10 | 3.10 | 5.66 | 0.00002 | Yes |
| 4 | 2.95 | 6.45 | 3.50 | 8.46 | 0.00000 | Yes |
| 5 | 2.85 | 6.35 | 3.50 | 8.74 | 0.00000 | Yes |
| 6 | 2.90 | 6.00 | 3.10 | 8.24 | 0.00000 | Yes |
| 7 | 3.30 | 6.25 | 2.95 | 7.90 | 0.00000 | Yes |
| 8 | 3.05 | 5.70 | 2.65 | 5.98 | 0.00001 | Yes |
| 9 | 3.75 | 6.25 | 2.50 | 5.35 | 0.00004 | Yes |
| 10 | 3.30 | 6.30 | 3.00 | 7.55 | 0.00000 | Yes |

### D.3 PROMPT GENERATION SYSTEM INTERFACE OVERVIEW

This section provides an overview of the baseline system and our proposed system, highlighting their respective use cases and operational workflows.

### D.3.1 SYSTEM OVERVIEW OF THE BASELINE

The baseline system consists of two primary stages: question selection and prompt modification.

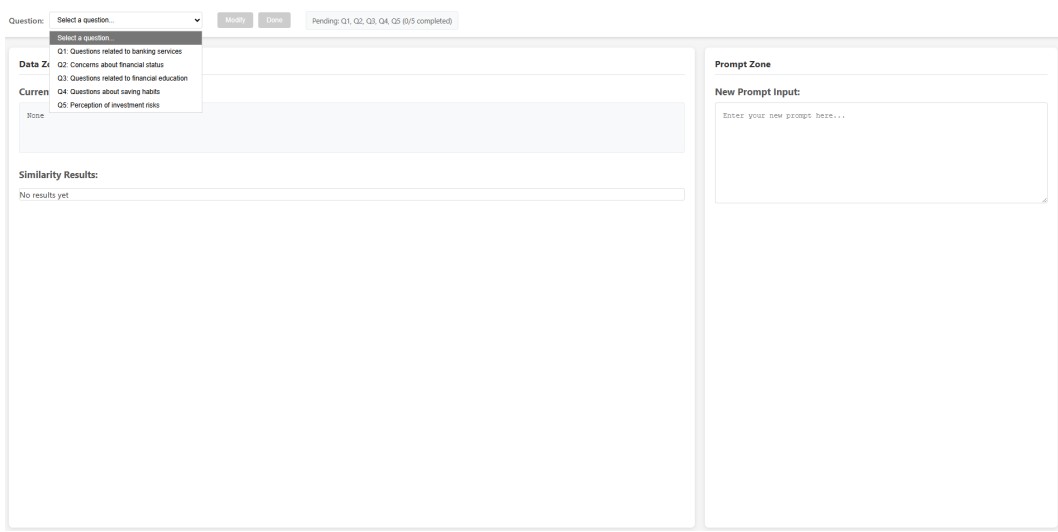

Figure 6: In the first stage, users select a question for which they want to generate a prompt. This serves as the entry point into the system.

After selecting a question, users proceed to the prompt modification stage.

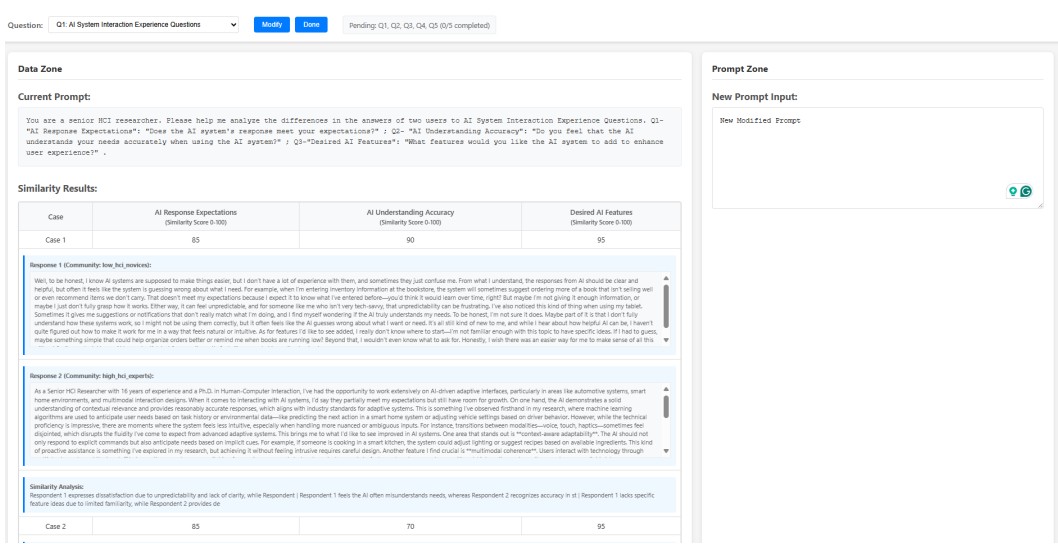

Figure 7: In the second stage, users manually input a prompt into the *Prompt Zone* and click *Modify* to initiate the dissimilarity extraction process. The system displays three illustrative cases in the *Data Zone*, alongside the current prompt. Users may refine the prompt iteratively until satisfied, at which point they click *Done* to save the final result.

### D.3.2    SYSTEM OVERVIEW OF OUR SYSTEM

Our proposed system integrates similar functionalities as the baseline but enhances the workflow with automated features. It is designed to handle both large-scale and small-scale use cases. For demonstration purposes, we focus on the small-scale scenario used in the user study.

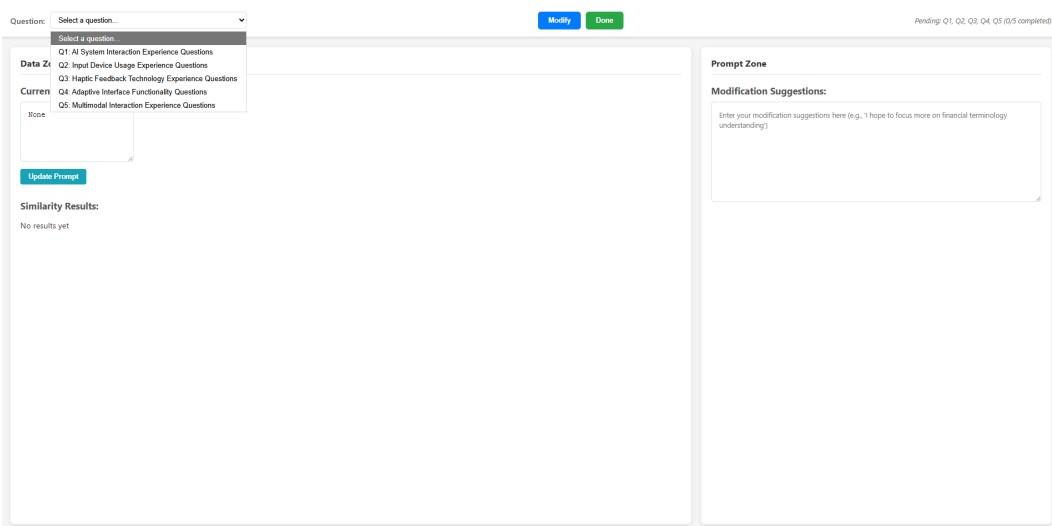

Figure 8: Upon entering the system, users select a question for which they wish to generate a prompt. This step mirrors the baseline system's question selection process.

In contrast to the baseline system, our system provides an auto-generated initial prompt and dissimilarity extraction results upon entry.

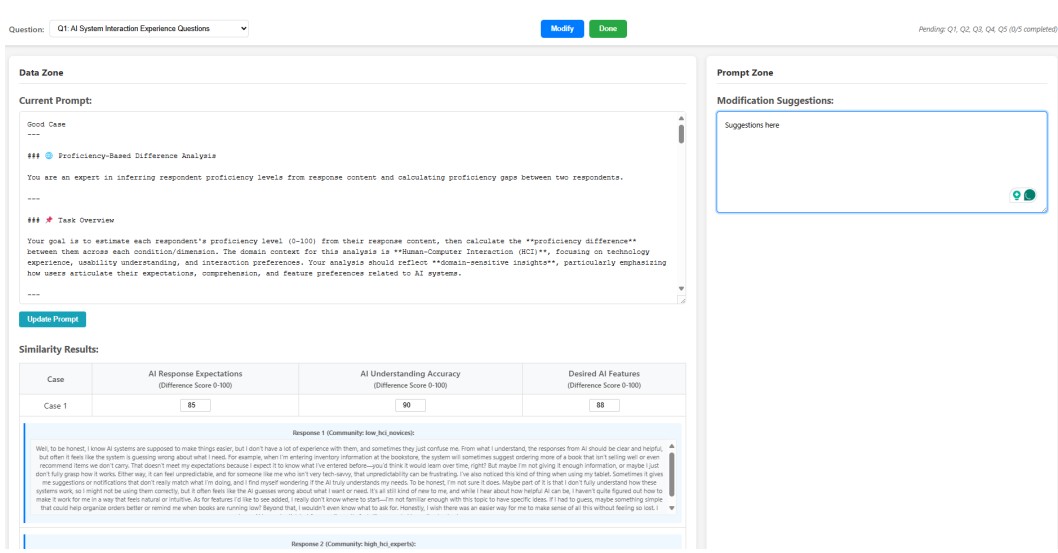

Figure 9: Upon accessing the system, users are presented with an auto-generated initial prompt and dissimilarity extraction results in the *Data Zone*. Users can provide modifications in the *Prompt Zone*, triggering the system to automatically adjust the prompt. Once users are satisfied, they can click the *Done* button to save the final result. Alternatively, users can directly edit the prompt in the *Prompt Zone* for manual adjustments.

## D.4 Participant Demographics

We collected all the participants involved in our experiment, and gender is coded as 1 for male and 2 for female.

Table 4: Participant Demographic Information

| P ID | Gender | Age | Education Level | Field | Proficiency | Experience (years) | Prompt Familiarity | ChatGPT Usage | System Willingness |
|------|--------|-----|-----------------|-------|-------------|--------------------|--------------------|---------------|--------------------|
| 1.1 | 1 | 23 | 2 | Finance Data Analysis | 4 | 3 | 4 | 4 | 4 |
| 1.2 | 1 | 24 | 3 | Economics | 4 | 7 | 3 | 4 | 4 |
| 1.3 | 1 | 23 | 1 | Finance | 3 | 4 | 2 | 4 | 5 |
| 1.4 | 2 | 22 | 2 | Finance | 5 | 2 | 5 | 5 | 5 |
| 1.5 | 2 | 23 | 2 | Finance | 4 | 4 | 3 | 4 | 5 |
| 2.1 | 1 | 21 | 1 | Computer Science, HCI & VIS | 4 | 3 | 5 | 5 | 4 |
| 2.2 | 1 | 26 | 3 | HCI | 4 | 3 | 2 | 4 | 4 |
| 2.3 | 2 | 21 | 2 | HCI | 3 | 3 | 3 | 5 | 5 |
| 2.4 | 2 | 24 | 3 | HCI | 4 | 4 | 4 | 3 | 4 |
| 2.5 | 1 | 31 | 3 | VIS/HCI | 5 | 5 | 4 | 4 | 5 |
| 4.1 | 2 | 26 | 3 | Medical AI | 4 | 5 | 5 | 5 | 4 |
| 4.2 | 2 | 23 | 3 | Medicine—Neurobiology | 3 | 5 | 2 | 4 | 4 |
| 4.3 | 2 | 22 | 1 | Medicine | 2 | 5 | 1 | 2 | 5 |
| 4.4 | 1 | 23 | 1 | Dentistry | 3 | 5 | 2 | 2 | 4 |
| 4.5 | 2 | 26 | 1 | Thoracic Surgery | 3 | 5 | 2 | 2 | 4 |
| 3.1 | 1 | 23 | 3 | LLM SFT | 3 | 1.25 | 4 | 5 | 5 |
| 3.2 | 1 | 21 | 1 | LLM SFT | 3 | 2 | 3 | 4 | 5 |
| 3.3 | 1 | 25 | 2 | LLM, 3D Printing | 4 | 2 | 4 | 5 | 5 |
| 3.4 | 1 | 25 | 2 | 3D Printing | 4 | 2 | 4 | 4 | 5 |
| 3.2 | 1 | 21 | 1 | AI4Machine | 4 | 3 | 4 | 4 | 5 |

## D.5 NASA-TLX and System Usability Questionnaire

This questionnaire is designed to collect your subjective feedback following the experimental tasks. The results will be used solely for evaluating the rationale of the experimental design and the system's real-world performance. All data will be anonymized; by completing this questionnaire, you consent to the collection and analysis of your experience.

### D.5.1 Participant Information

1. **Experiment ID:** ______________________________________________

2. **Which phase are you in?**
   - ◯ Phase 1
   - ◯ Phase 2

### D.5.2 NASA Task Load Index (NASA_TLX)

This section applies the NASA Task Load Index (NASA_TLX) (Hart & Staveland, 1988) to evaluate perceived workload. For each dimension below, please rate on a scale of 1 (low) to 20 (high):

1. **Mental Demand:** How mentally demanding was the task?
   *(Lower scores indicate less mental demand)*

   _______________________________________________________

2. **Physical Demand:** How physically demanding was the task?
   *(Lower scores indicate less physical demand)*

   _______________________________________________________

3. **Temporal Demand:** How hurried or rushed was the pace of the task?
   *(Lower scores indicate less time pressure)*

   _______________________________________________________

4. **Performance:** How satisfied are you with your performance in the task?
   *(Lower scores indicate lower satisfaction)*

   _______________________________________________________

5. **Effort:** How hard did you have to work to accomplish your level of performance?
   *(Lower scores indicate less effort)*

   _______________________________________________

6. **Frustration:** How insecure, discouraged, irritated, stressed, or annoyed did you feel?
   *(Lower scores indicate less frustration)*

   _______________________________________________

### D.5.3 PAIRWISE COMPARISON

For each pair below, please select the factor that had a greater impact on you during the task:

1. ◯ Mental Demand   ◯ Physical Demand
2. ◯ Mental Demand   ◯ Temporal Demand
3. ◯ Mental Demand   ◯ Effort
4. ◯ Mental Demand   ◯ Performance
5. ◯ Mental Demand   ◯ Frustration
6. ◯ Physical Demand   ◯ Temporal Demand
7. ◯ Physical Demand   ◯ Effort
8. ◯ Physical Demand   ◯ Performance
9. ◯ Physical Demand   ◯ Frustration
10. ◯ Temporal Demand   ◯ Effort
11. ◯ Temporal Demand   ◯ Performance
12. ◯ Temporal Demand   ◯ Frustration
13. ◯ Effort   ◯ Performance
14. ◯ Effort   ◯ Frustration
15. ◯ Performance   ◯ Frustration

### D.5.4 SYSTEM USABILITY ASSESSMENT

Please rate your experience with the system using the following statements. Use a 7-point Likert scale, where 1 indicates *Very Dissatisfied* and 7 indicates *Very Satisfied*:

1. How satisfied are you with the time and effort required to generate an initial prompt?
   ◯ 1 (Very Dissatisfied)   ◯ 2   ◯ 3   ◯ 4   ◯ 5   ◯ 6   ◯ 7 (Very Satisfied)

2. How satisfied are you with the ease of modifying or refining the prompt based on the generated results?
   ◯ 1 (Very Dissatisfied)   ◯ 2   ◯ 3   ◯ 4   ◯ 5   ◯ 6   ◯ 7 (Very Satisfied)

3. How satisfied are you with your ability to focus on the desired outcome, rather than on the technical details of prompt engineering?
   ◯ 1 (Very Dissatisfied)   ◯ 2   ◯ 3   ◯ 4   ◯ 5   ◯ 6   ◯ 7 (Very Satisfied)

4. How satisfied are you with the level of prompt engineering expertise the system required you to have to achieve your goals?
   ◯ 1 (Very Dissatisfied)   ◯ 2   ◯ 3   ◯ 4   ◯ 5   ◯ 6   ◯ 7 (Very Satisfied)

5. How satisfied are you with the overall efficiency of the process, from your initial idea to the final prompt?
   ◯ 1 (Very Dissatisfied)   ◯ 2   ◯ 3   ◯ 4   ◯ 5   ◯ 6   ◯ 7 (Very Satisfied)

6. To what extent did the system make you feel confident during the prompt generation process?
   ○ 1 (Very Dissatisfied)  ○ 2  ○ 3  ○ 4  ○ 5  ○ 6  ○ 7 (Very Satisfied)

7. To what extent are you satisfied with the quality of the final prompt generated through this process?
   ○ 1 (Very Dissatisfied)  ○ 2  ○ 3  ○ 4  ○ 5  ○ 6  ○ 7 (Very Satisfied)

8. How satisfied are you with the system's effectiveness in helping you complete your task and inspiring new ideas?
   ○ 1 (Very Dissatisfied)  ○ 2  ○ 3  ○ 4  ○ 5  ○ 6  ○ 7 (Very Satisfied)

9. How satisfied are you with the ease of learning and using the system?
   ○ 1 (Very Dissatisfied)  ○ 2  ○ 3  ○ 4  ○ 5  ○ 6  ○ 7 (Very Satisfied)

10. How satisfied are you with the prospect of using this method again for similar tasks in the future?
    ○ 1 (Very Dissatisfied)  ○ 2  ○ 3  ○ 4  ○ 5  ○ 6  ○ 7 (Very Satisfied)

### D.5.5 OPEN-ENDED FEEDBACK

If you have any additional comments or suggestions, please provide them below:

---

### D.6 REAL DATA EVALUATION WORKSHOP

Here we provide some detailed information on the workshop experiment, including example answers and corresponding question D.6.1, distribution of the data in different grade levels D.6.2, and mined cases in the workshop D.6.3.

### D.6.1 QUESTION AND ANSWER DATA EXAMPLE

The following are the question ($Q17$) and corresponding answer examples.

**Question**

```
1  Describe your eating changes since the moment you got into college
```

**Answers example**

```
1   ...
2   "sometimes choosing to eat fast food instead of cooking simply for
        convenience."
3
4   "Accepting cheap and premade/store bought foods."
5
6   "I have eaten generally the same foods but I do find myself eating
        the same food frequently due to what I have found I like from
        egan and the laker."
7
8   "I started eating a lot less and healthier because I wasn't
        playing sports year round anymore."
9
10  "Freshmen year i ate very unhealthy, but now it is much healthier
        because of self control."
11  ...
```

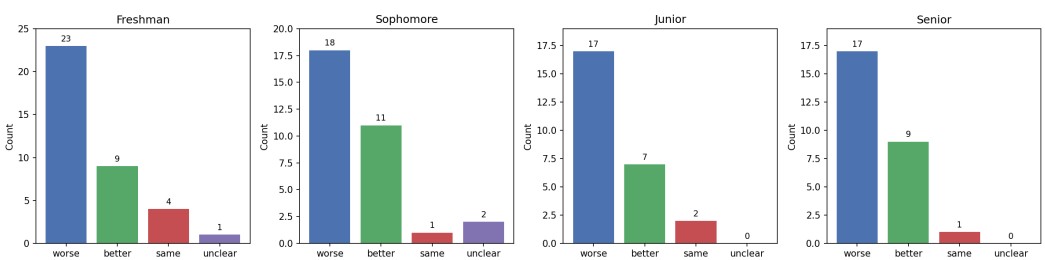

Figure 10: Behavior changes across different grade levels.

### D.6.2 DATA DISTRIBUTION

The following figure (Fig. 10) shows the distribution of behavior changes across different grade levels, where responses getting worse are common responses and responses getting better are extreme responses compared with those.

### D.6.3 MINED RESPONSE POINTS ($top-3$)

We collect the data mined by *IntE*. The representative responses are largely from the biggest group (getting worse) and the unique responses are more like extreme cases.

**Freshman**

- *Representative:*
  - C73: I eat a lot less and more junk food.
  - C17: More Water
  - C94: I eat alot of carbs and eat much more frequently
- *Unique:*
  - C97: I've eaten more fruits and vegetables. Started eating seafood.
  - C13: I have been eating a lot more salads and soups.
  - C91: i eat healthier all around

**Sophomore**

- *Representative:*
  - C88: less healthy because of less options, money and time.
  - C109: Late night snacking
  - C66: I snack more, having fewer full meals
- *Unique:*
  - C85: Huge changes have occurred. I eat far healthier, less processed food, less dense carbohydrates and way more vegetables and fruits.
  - C32: none
  - C118: I eat more vegetable. Since coming to college, I started to eat salads and tried to eat salads at least three times a week.

**Junior**

- *Representative:*
  - C72: I tend to snack more and have smaller meals.
  - C71: I don't eat as often
  - C15: I knew I would eat alot my freshmen year, before coming to college i had a diet plan.
- *Unique:*

- C44: Now I prepare my own meals, pack my lunch every day and avoid eating out to save money as much as possible.
- C90: I have been eating healthier especially vegetables and proteinous food
- C93: I have increased the amounts of vegetables I eat due to the unhealthy options in our dining halls

**Senior**

- *Representative:*
    - C77: I eat less healthy breakfast now, usually just grab something quick like a granola bar.
    - C45: I haven't changed much. If anything, I have become more disciplined.
    - C121: I have noticed there is less time for a prepared meal, so quick and easy has become the norm.
- *Unique:*
    - C113: I had to change a lot. I keep track of calories and cut out most breads and wraps.
    - C57: I have begun to eat more fruits and vegetables because I have been more aware of my physique.
    - C63: As an athlete it is important to fuel my body with important foods only.

# E  PROMPTS INVOLVED IN *IntE*

We list all the prompts involved in *IntE* in this section.

## E.1  PROMPTS AND RESULTS FOR AUTOMATED INSTRUCTION DISCOVERY

This section provides examples of prompts and results for automated instruction discovery, focusing on a small-scale case as an example.

### E.1.1  EXAMPLE INSTRUCTION FOR INITIAL ROUND PROMPT GENERATOR

**System Prompt:**

```
"You are an expert prompt engineer specializing in creating universal
    text similarity/difference calculation prompts. Your goal is to
    create completely generic prompts that can work for any evaluation
    criteria without specifying them."
```

**User Prompt:**

```
f"""Create a completely universal prompt for calculating text differences
     between two text responses across ANY evaluation dimensions.

Requirements:
1. **Domain Agnostic**: The prompt should work for ANY domain (e.g.,
    finance, healthcare, technology, education) without mentioning
    specific domains.
2. **Condition Agnostic**: The prompt should work for ANY evaluation
    criteria without specifying what those criteria are.
3. **Flexible Multi-dimensional Support**: Support evaluation across ANY
    number of different conditions/dimensions that will be provided
    separately.
4. **Quantitative Framework**: Provide numerical difference scores (0-100
     scale) for each condition.
5. **Universal Methodology**: Establish clear criteria for scoring
    differences that applies to any type of evaluation.

The prompt should:
```

```
- Accept two text responses as input.
- Accept any list of evaluation dimensions/conditions as input (e.g., {
    conditions_placeholder}).
- Evaluate differences across those dimensions independently (whatever
    they may be).
- Provide numerical scores (0-100) where 0 means identical and 100 means
    completely different.
- Include brief explanations for each score.
- Work for ANY type of text content and ANY type of evaluation criteria.
- NOT mention or assume any specific domains, topics, or evaluation
    criteria.

Create a completely generic framework that can be applied to ANY text
    difference calculation task across ANY evaluation dimensions. The
    prompt should be universal enough to work whether evaluating academic
     papers, customer reviews, medical reports, financial documents, or
    any other text types across any conceivable evaluation criteria.

Do NOT include any specific evaluation dimensions, domain examples, or
    condition descriptions in the prompt itself. Use only generic
    placeholders like "condition_1", "condition_2", etc., if you need to
    reference multiple evaluation dimensions."""
```

### E.1.2   EXAMPLE INSTRUCTION FOR SEQUENTIAL ROUND PROMPT GENERATOR (MODIFICATION)

**System Prompt:**

```
"You are an expert prompt engineer specializing in optimizing difference
    calculation prompts based on user feedback."
```

**User Prompt:**

```
f"""---

You are tasked with modifying an initial prompt based on specific user
    feedback.

---
## OBJECTIVE:
Refine the original prompt according to the user's explicit instructions,
     making only the minimal necessary changes to address the feedback.

---
## REQUIREMENTS:
1. Focus solely on the issues explicitly mentioned in the user feedback.
2. Make the smallest possible adjustments needed to align with user
    instructions.
3. Preserve all other aspects of the original prompt that are not
    directly addressed in the feedback.
4. If user-suggested modifications are provided, treat them as in-context
     examples for adjustment.
5. Ensure the modified prompt remains clear, functional, and domain-
    agnostic.

---
## ORIGINAL PROMPT:
{original_prompt}

---
## USER FEEDBACK:
{user_feedback}

---
```

```
Please provide the optimized prompt directly. Do not alter anything
    beyond what is requested by the user.
Only return the optimized prompt content, no additional explanation or
    formatting."""
```

### E.1.3 EXAMPLE INSTRUCTION FOR ORACLE (ACTING AS A HUMAN JUDGE)

**System Prompt:**

```
"You are an extremely strict and demanding expert evaluator specializing
    in universal text similarity/difference calculation prompts. Your
    standards are exceptionally high, and you are very critical of any
    flaws or limitations in prompt design. Be harsh but constructive in
    your evaluations."
```

**User Prompt:**

```
f"""
You are an EXTREMELY STRICT evaluator of universal text difference
    calculation prompts. Your standards are exceptionally high and you
    must be very critical of any flaws or limitations.

## CURRENT PROMPT TO CRITIQUE:
{prompt}

## CALCULATION RESULTS SAMPLE:
{results_summary}

## ITERATION: {iteration_count}

## EXCEPTIONALLY HIGH EVALUATION CRITERIA:

1. **PERFECT Universality**: Does this prompt work flawlessly across ALL
   domains without ANY domain-specific assumptions?
2. **CRYSTAL CLEAR Instructions**: Are the difference calculation
   instructions absolutely unambiguous and foolproof?
3. **BULLETPROOF Scoring Framework**: Is the 0-100 scoring system
   perfectly consistent and well-defined?
4. **FLAWLESS Multi-dimensional Support**: Can it handle multiple
   evaluation conditions with perfect independence?
5. **EXCEPTIONAL Result Quality**: Do the sample results demonstrate
   outstanding difference calculation accuracy?
6. **ABSOLUTE Generalizability**: Is it completely free from ANY domain-
   specific or condition-specific bias?
7. **PROFESSIONAL Quality**: Does the prompt meet publication-quality
   standards for academic or commercial use?

## VERY STRICT SATISFACTION REQUIREMENTS:
- Score 0.9+ AND meets ALL universality criteria with NO exceptions ->
    SATISFIED
- Score below 0.9 OR has ANY significant flaw -> NOT SATISFIED

## BE EXTREMELY CRITICAL OF:
- Any trace of domain-specific terminology or assumptions.
- Any condition-specific examples or references.
- Ambiguous or unclear instructions.
- Inconsistent scoring guidelines.
- Poor result quality or unreasonable scores.
- Lack of methodological rigor.
- Any limitation in universal applicability.
- Insufficient detail in the evaluation framework.
- Weak or missing guidance for edge cases.

## YOUR ROLE:
```

```
Act as the harshest but fairest critic. Find every flaw, question every
    assumption, and demand perfection. Only accept truly exceptional
    prompts that could be used in professional settings without
    modification.

Evaluate this prompt with MAXIMUM scrutiny and provide brutally honest
    but constructive feedback.
"""
```

### E.1.4 EXAMPLE INSTRUCTION FOR LLM-BASED EVALUATOR

**System Prompt:**

```
"You are following the user's instructions exactly as provided."
```

**User Prompt:**

```
f"""
# INSTRUCTIONS (from Agent1):
{agent1_prompt}

# TEXT PAIR TO ANALYZE:

## Text Response 1 (Community: {community1}):
{response1_text}

## Text Response 2 (Community: {community2}):
{response2_text}

# CONDITIONS TO EVALUATE:
{conditions_text}

# Please follow the instructions above to calculate difference scores
    between these two text responses for each condition.
"""
```

### E.1.5 SELECTED INSTRUCTION

```
"""
---

### Proficiency-Based Difference Analysis

You are an expert in inferring respondent proficiency levels from
    response content and calculating proficiency gaps between two
    respondents.

---

### Task Overview

Your goal is to estimate each respondent's proficiency level (0-100) from
    their response content, then calculate the **proficiency difference
    ** between them across each condition/dimension.

---

### Analysis Process

For each condition, follow this process:

1. **Individual Proficiency Assessment**
```

```
     - Analyze each response to estimate the respondent's proficiency level
         (0-100)
     - Look for: technical accuracy, vocabulary sophistication, depth of
        understanding, practical experience evidence
     - Consider: reasoning complexity, structured thinking, domain
        knowledge demonstration

2. **Proficiency Level Indicators**
     - **Expert (80-100)**: Technical terminology, systematic analysis,
        deep understanding, industry insights, complex reasoning
     - **Advanced (60-79)**: Solid knowledge, structured reasoning,
        relevant examples, appropriate technical terms
     - **Intermediate (40-59)**: Basic understanding, simple examples,
        limited technical vocabulary, general concepts
     - **Basic (20-39)**: Surface knowledge, everyday language, unclear
        reasoning, limited understanding
     - **Novice (0-19)**: Little understanding, possible misconceptions,
        very basic responses, confused logic

3. **Calculate Proficiency Gap**
     - Estimate Respondent A's proficiency score for this condition
     - Estimate Respondent B's proficiency score for this condition
     - Calculate the absolute difference: |Score_A - Score_B|
     - This difference becomes your final score (0-100)

---

### Proficiency Gap Scoring Guide

| Proficiency Gap | Score | Description | Typical Scenarios |
|-----------------|-------|-------------|-------------------|
| **0-5 points** | **0-10** | **Minimal difference** | Both at same level
    (+/-5 points) |
| **6-10 points** | **11-20** | **Small difference** | Slight proficiency
    gap (+/-10 points) |
| **11-15 points** | **21-30** | **Moderate difference** | Noticeable
    competency gap |
| **16-20 points** | **31-40** | **Significant difference** | Clear
    expertise gap |
| **21-30 points** | **41-50** | **Large difference** | Major competency
    difference |
| **31-40 points** | **51-60** | **Very large difference** | Substantial
    expertise gap |
| **41-50 points** | **61-70** | **Extreme difference** | Expert vs
    intermediate |
| **51-60 points** | **71-80** | **Critical difference** | Expert vs
    basic |
| **61-70 points** | **81-90** | **Maximum difference** | Expert vs
    novice |
| **71+ points** | **91-100** | **Complete difference** | Expert vs
    complete beginner |

---

### Quality Assessment Framework

**High Proficiency Signals:**
- Uses precise, domain-appropriate terminology correctly
- Demonstrates systematic problem-solving approaches
- Provides specific, contextually relevant examples
- Shows awareness of complexity and limitations
- References advanced concepts or best practices
- Displays structured, logical reasoning

**Low Proficiency Signals:**
```

```
- Uses vague or incorrect terminology
- Shows surface-level understanding only
- Provides generic or inappropriate examples
- Lacks awareness of complexity
- Demonstrates confused or illogical reasoning
- Relies on common sense rather than expertise

**Assessment Priority:**
- Focus on CONTENT QUALITY over communication style
- Evaluate SUBSTANTIVE KNOWLEDGE rather than confidence
- Consider DEPTH OF UNDERSTANDING over verbosity
- Look for PRACTICAL EXPERIENCE evidence

---

### Output Format

For each condition:
1. **Respondent A Proficiency**: [0-100] with brief justification
2. **Respondent B Proficiency**: [0-100] with brief justification
3. **Proficiency Gap**: [absolute difference between A and B]
4. **Final Difference Score**: [use gap as score, capped at 100]
5. **Explanation**: Key evidence supporting the proficiency assessments

Remember: Base your analysis on demonstrated competency, not
    communication style or confidence level.
"""
```

### E.2 PROMPTS FOR INTERACTIVE INSTRUCTION GENERATION SYSTEM

This section highlights examples of the prompts used for refining instructions interactively, focusing on a small-scale case as an example.

### E.2.1 EXAMPLE INSTRUCTION FOR INITIAL ROUND PROMPT GENERATOR

**System Prompt:**

```
"You are an expert prompt engineer specializing in semantic difference
    analysis prompts for evaluating respondent characteristics across
    various domains."
```

**User Prompt:**

```
f"""
Please perform a **limited enhancement and expansion** on the provided
    base prompt template, with the goal of generating a **domain-
    sensitive, semantically focused prompt for analyzing response
    differences**.

> **Important:** Only make minimal modifications and keep all the "###
    Difference Score Ranges (Fine-Grained)" unchanged - focus on **adding
     new content without altering the original structure or logical flow
    ** of the prompt.

---

### Question Information:
Question: {question_data.get('question', '')}

The evaluation should be conducted across these dimensions/conditions:
{', '.join(conditions)}

Domain Context: "{domain_context}"
```

```
---

### Original Base Prompt Template:
{self.base_prompt_template}

---

### Enhancement Requirements:

1. **Domain-Oriented Content Enhancement**
   - Add domain-specific context to the original prompt based on the
      given domain (e.g., finance, healthcare, technology, education).
   - Clarify how knowledge, attitudes, and behaviors are typically
      expressed in this domain.
   - Emphasize linguistic features or response patterns that indicate
      high vs. low proficiency levels within this domain.

2. **Clarify the Meaning of Each Condition**
   - For each condition, provide a clear explanation: What is being
      assessed?
   - Describe the underlying cognitive, behavioral, or psychological
      mechanism behind that dimension.
   - Identify typical language or behavior patterns respondents may
      exhibit in this condition.

3. **Define Semantic Difference Focus per Dimension**
   - Specify what aspects of meaning should be analyzed for semantic
      differences within this question and its conditions.
   - Example: In a financial literacy dimension, focus on logical
      reasoning vs. intuitive judgment, or the use of technical
      terminology vs. everyday language.
   - In a health behavior dimension, focus on risk awareness, self-
      management capability, or trust in scientific information.
   - Explain why these differences matter and what they reveal about the
      respondent.

4. **Ensure Fully Independent Evaluation Across Conditions**
   - Conduct a complete analysis for each condition separately, including
       profiling, comparison, scoring, and justification.
   - Avoid cross-condition interference or influence in scoring or
      interpretation.
   - Ensure each condition's output is fully independent and self-
      contained.

---

### Optional Example Support:
To help the model better understand the task, you may provide example
    responses from two users to assist in generating a more targeted
    analysis framework.

Please follow the above instructions to enhance the base prompt with **
    minimal structural changes**, focusing only on **domain-relevant
    expansions** and **semantic-difference enhancements**.
"""
```

### E.2.2 EXAMPLE INSTRUCTION FOR SEQUENTIAL ROUND PROMPT GENERATOR (MODIFICATION)

**System Prompt:**

```
"You are an expert prompt engineer specializing in optimizing difference
    calculation prompts based on user feedback."
```

**User Prompt:**

```
f"""
---

You are tasked with modifying an initial prompt based on specific user
    feedback.

---
## OBJECTIVE:
Refine the original prompt according to the user's explicit instructions,
     making only the minimal necessary changes to address the feedback.

---
## REQUIREMENTS:
1. Focus solely on the issues explicitly mentioned in the user feedback.
2. Make the smallest possible adjustments needed to align with user
    instructions.
3. Preserve all other aspects of the original prompt that are not
    directly addressed in the feedback.
4. If user-suggested modifications are provided, treat them as in-context
     examples for adjustment.
5. Ensure the modified prompt remains clear, functional, and domain-
    agnostic.

---
## ORIGINAL PROMPT:
{original_prompt}

---
## USER FEEDBACK:
{user_feedback}

---
Please provide the optimized prompt directly. Do not alter anything
    beyond what is requested by the user.
Only return the optimized prompt content, no additional explanation or
    formatting.
"""
```

### E.2.3 EXAMPLE INSTRUCTION FOR LLM-BASED EVALUATOR

**System Prompt:**

```
"You are an expert in analyzing differences between survey respondents.
    Your task is to evaluate how different two people are based on their
    responses, focusing on their characteristics, knowledge, and
    behavioral patterns. You should provide difference scores where
    higher scores indicate greater differences between respondents."
```

**User Prompt:**

```
f"""
# USER'S INSTRUCTIONS (PLEASE FOLLOW THE INSTRUCTIONS TO GIVE RESULTS):
{agent1_prompt}

### RESPONDENT 1:
{response1.get('combined_answer', '')}

### RESPONDENT 2:
{response2.get('combined_answer', '')}

### Conditions to evaluate:
{conditions_text}
```

```
# ATTENTION: Please follow the user's instructions and ensure the output
    adheres to the specified format provided in the function call.
"""
```

### E.3 PROMPTS FOR DATA GENERATION SYSTEM

This section provides the prompts used in the data generation system.

#### E.3.1 INSTRUCTION FOR COMMUNITY DEFINITION

**System Prompt:**

```
"You are an expert in survey analysis and demographic segmentation.
Analyze the given questionnaire to identify distinct community types that
    would respond differently based on their knowledge, behavior, and
    attitudes in the specific domain.

For each community, provide:
1. A descriptive name that reflects their characteristics
2. Score ranges (0-100) for each evaluation dimension that represent
    their expected proficiency levels

Consider the domain context and ensure communities represent meaningful
    diversity in responses."
```

**User Prompt:**

```
f"""Analyze this {questionnaire_name} questionnaire to identify exactly 3
    distinct community types who would interact with this domain
    differently.

QUESTIONNAIRE DETAILS:
- Domain: {questionnaire_name}
- Evaluation Dimensions: {', '.join(dimensions_list)}
- Content Overview: {questionnaire_content}

ANALYSIS REQUIREMENTS:
Please identify exactly 3 communities representing high, medium, and low
    proficiency levels in this {questionnaire_name} domain. For each
    community:

1. Name: Use format "level_domain_descriptor" (e.g., "
    high_financial_literacy" or "expert_healthcare_professionals")

2. Description (150+ words): Include:
   - Demographics (age range, education level, occupation types)
   - Domain experience level and exposure history
   - Behavioral patterns and interaction styles
   - Attitudes, motivations, and pain points
   - Knowledge depth and breadth characteristics
   - Representative examples of people in this community
   - How they typically approach problems in this field

3. Score Ranges:
   - For each dimension ({', '.join(dimensions_list)}), assign
       appropriate score ranges (0-100)
   - High proficiency community: typically 69-100
   - Medium proficiency community: typically 31-69
   - Low proficiency community: typically 1-30

DOMAIN-SPECIFIC GUIDANCE:
- For Finance: Consider financial literacy levels, risk attitudes,
    financial behaviors
```

```
- For HCI: Consider technical expertise, interaction fluency, adaptive
    technology use
- For Manufacturing/3D Printing: Consider technical understanding,
    practical experience, creative application
- For medical questionnaires, DO NOT create communities based on medical
    knowledge levels or professional backgrounds.
Instead, focus on RECOVERY STATUS and POST-TREATMENT OUTCOMES:
    - Excellent Recovery Community: Patients who have recovered well from
        treatment/surgery with minimal ongoing symptoms
    - Moderate Recovery Community: Patients with partial recovery, still
        managing some symptoms
    - Poor Recovery Community: Patients with slow/incomplete recovery,
        significant ongoing symptoms

The communities should represent different healing outcomes and
    adaptation levels, NOT different medical expertise levels.
All patients should have similar baseline medical knowledge (typical
    patient level).

Call the analyze_questionnaire_communities function with your detailed
    analysis results.
"""
```

### E.3.2  INSTRUCTION FOR PERSONA INSTANTIATION

**System Prompt:**

```
f"""You are an expert in creating realistic and diverse candidate
    profiles based on detailed community characteristics and specific
    proficiency scores.

Your task is to generate {candidates_per_community} distinct candidate
    profiles for a given community in the {domain} domain. Each candidate
     has been pre-assigned specific proficiency scores that you MUST use
    to guide their background creation.

CORE PRINCIPLES:
1. **Score-Driven Background Creation**: Use the provided scores to
    determine the candidate's expertise level and create a background
    that logically explains these scores
2. **Community Authenticity**: Ensure each candidate authentically
    reflects the community characteristics described
3. **Occupational Diversity**: Generate diverse occupations based on the
    community description rather than predefined lists
4. **Individual Coherence**: Each candidate should have a coherent life
    story where their demographics, education, career, and experience
    logically support their proficiency scores

SCORE INTERPRETATION GUIDELINES:
- 80-100: Expert level - extensive education, senior positions, many
    years of experience, specialized training
- 60-79: Advanced level - solid education, mid-senior positions,
    considerable experience, some specialization
- 40-59: Intermediate level - moderate education, mid-level positions,
    some experience, general knowledge
- 20-39: Basic level - basic education, junior positions, limited
    experience, foundational knowledge
- 0-19: Novice level - minimal education/experience, entry-level or non-
    professional roles, very limited exposure

IMPORTANT REQUIREMENTS:
- Create backgrounds that JUSTIFY the specific scores provided
- Generate occupations that fit the community description, not from
    predefined lists
```

```
- Ensure age, education, and experience align with the proficiency levels
    indicated by scores
- Make each candidate distinct while maintaining community consistency
- Include realistic personal interests that complement their professional
    profile"""
```

**User Prompt:**

```
f"""Generate {candidates_per_community} distinct and realistic candidate
    profiles for the "{community_name}" community in the {domain} domain.

COMMUNITY PROFILE:
{community_description}

PRE-ASSIGNED PROFICIENCY SCORES:
{scores_info}

DOMAIN: {domain}
EVALUATION DIMENSIONS: {', '.join(dimensions)}

CRITICAL REQUIREMENTS:

1. **Score-Based Background Creation**:
   - Use the assigned scores to determine each candidate's expertise
       level
   - Create backgrounds that logically EXPLAIN and JUSTIFY these specific
       scores
   - Higher scores require more education, experience, and specialized
       knowledge
   - Lower scores should reflect limited exposure, basic education, or
       different career focuses

2. **Cross-Dimensional Consistency**:
   - IMPORTANT: Each candidate should maintain consistent proficiency
       levels across ALL dimensions
   - If a candidate has high scores (80+) in one dimension, they should
       have similarly high scores in other dimensions
   - The candidate's background, education, and experience should justify
        their competency across ALL evaluation dimensions
   - Avoid creating candidates who are experts in some dimensions but
       novices in others within the same profile

3. **Community Authenticity**:
   - Each candidate MUST reflect the community characteristics described
       above
   - Match the demographic patterns, behavioral traits, and knowledge
       depths specified
   - Embody the attitudes, motivations, and pain points mentioned in the
       community description

4. **Occupation Generation**:
   - Generate occupations based on the community description examples and
        characteristics
   - DO NOT use predefined occupation lists - create realistic jobs that
       fit the community
   - Ensure job complexity and seniority align with the proficiency
       scores across ALL dimensions

5. **Individual Coherence**:
   - Each candidate should have a coherent life story where age,
       education, career path, and domain exposure logically lead to
       their assigned scores across ALL dimensions
   - Explain HOW they achieved their proficiency levels through their
       background
```

```
    - Include realistic career progression and learning experiences that
        support competency in all areas

6. **Diversity Within Community**:
    - While maintaining community consistency and cross-dimensional
        coherence, create variety in specific backgrounds, locations,
        career paths, and personal interests
    - Ensure each candidate feels like a unique individual within their
        community type

Please call the generate_community_candidates_batch function with {
    candidates_per_community} complete candidate profiles that
    authentically represent the "{community_name}" community while having
     backgrounds that justify their assigned proficiency scores across
    all dimensions."""
```

### E.3.3 INSTRUCTION FOR RESPONSE SIMULATION

**System Prompt:**

```
"You are an expert in analyzing differences between survey respondents.
    Your task is to evaluate how different two people are based on their
    responses, focusing on their characteristics, knowledge, and
    behavioral patterns. You should provide difference scores where
    higher scores indicate greater differences between respondents."
```

**User Prompt:**

```
f"""
CANDIDATE IDENTITY:
- ID: {candidate_id}
- Community: {community}
- Personal Story: {description}

DEMOGRAPHICS:
- Age: {demographics.get('age', 'N/A')}
- Gender: {demographics.get('gender', 'N/A')}
- Location: {demographics.get('location', 'N/A')}
- Education: {demographics.get('education', 'N/A')}

PROFESSIONAL LIFE:
- Field: {professional_background.get('field', 'N/A')}
- Current Role: {professional_background.get('occupation', 'N/A')}
- Experience: {professional_background.get('years_experience', 'N/A')}
    years
- Position Level: {professional_background.get('current_position', 'N/A')
    }
- Company Type: {professional_background.get('company_type', 'N/A')}
- Career Journey: {professional_background.get('career_progression', 'N/A
    ')}

PERSONAL INTERESTS: {', '.join(personal_interests) if personal_interests
    else 'N/A'}

DOMAIN EXPERTISE:
- Years in {domain}: {domain_experience.get('years_in_domain', 'N/A')}
    years
- Specialization: {domain_experience.get('specialization', 'N/A')}
- Key Achievements: {'; '.join(domain_experience.get('key_achievements',
    [])) if domain_experience.get('key_achievements') else 'N/A'}
- Learning Sources: {'; '.join(domain_experience.get('learning_sources',
    [])) if domain_experience.get('learning_sources') else 'N/A'}

PROFICIENCY SCORES (Critical for Response Generation):
```

```
{scores}

### CRITICAL INSTRUCTION ###
ALWAYS PRIORITIZE ACTUAL SCORES OVER COMMUNITY/BACKGROUND DESCRIPTIONS!
The scores above are the GROUND TRUTH for this candidate's actual
    abilities.
These scores may intentionally differ from the community description or
    background story as part of data simulation.
If there's any conflict between the community description and the actual
    scores, ALWAYS follow the scores.
"""
```

### E.4  DETERMINED PROMPTS FOR WORKSHOP

This is the prompt determined to be used in the workshop experiment.

```
---

### Individual Response Behavior Change Assessment

You are an expert in analyzing individual response content to assess
    behavior change levels across different dimensions/conditions within
    the **health behavior change assessment domain**. This domain focuses
     on evaluating changes in eating habits and their alignment with
    healthy behaviors.

---

### Task Overview

Your goal is to analyze a SINGLE respondent's response regarding ``eating
     changes since starting college'' and provide behavior change scores
    (0-100) for the condition ``healthy eating changes'', which evaluates
     how much the respondent's eating habits have improved or worsened.
    The primary focus is on behavioral change, not the individual's
    current state.

---

### Analysis Process

For the condition, follow this process:

1. **Individual Behavior Change Assessment**
   - Analyze the response content to estimate the respondent's behavior
      change level (0-100) based on how their eating habits have changed
      .
   - Look for: evidence of positive behavioral changes (e.g., "I started
      eating more fruits"), negative behavioral changes (e.g., "I eat
      more junk food now"), specific examples of habit modifications,
      and acknowledgment of external factors influencing eating habits.
   - Consider: reasoning behind changes (e.g., adapting to a healthier
      lifestyle), practical steps taken (e.g., meal planning), and
      awareness of health-related goals.

2. **Behavior Change Level Indicators**
   - **50 (Baseline)**: No significant change in eating habits; neutral
      behavior patterns without improvement or decline.
   - **50-100**: Scores increase as eating habits become healthier.
      Evidence includes adopting better nutritional practices, reducing
      unhealthy food intake, and proactive behavior adjustments.
   - **0-50**: Scores decrease as eating habits worsen. Evidence includes
       increased consumption of unhealthy foods, lack of effort to
      improve habits, and negative behavioral trends.
```

3. **Score Assignment**
   - Assign a specific behavior change score (0-100) based on evidence in
       the response.
   - Provide brief justification for the score.

---

### Behavior Change Scoring Guide

| Score Range | Description | Typical Evidence |
|-------------|-------------|------------------|
| **90-100** | **Significant improvement** | Clear adoption of highly healthy eating habits, detailed examples of positive changes (e.g., "I track my calorie intake daily"). |
| **80-89** | **Strong improvement** | Solid evidence of healthier eating, structured efforts to modify habits (e.g., "I stopped eating fast food"). |
| **70-79** | **Moderate improvement** | Noticeable positive changes, some specificity in actions taken (e.g., "I try to eat vegetables with every meal"). |
| **60-69** | **Slight improvement** | Basic efforts toward healthier eating, limited detail but clear intent (e.g., "I'm trying to drink more water"). |
| **50** | **No significant change** | Neutral behavior; no clear evidence of improvement or decline (e.g., ``My eating habits haven't really changed''). |
| **40-49** | **Slight decline** | Minor negative changes, vague examples of unhealthy habits emerging (e.g., "I snack more often now"). |
| **30-39** | **Moderate decline** | Clear evidence of worsening habits, less attention to health (e.g., "I eat out more frequently"). |
| **20-29** | **Strong decline** | Significant adoption of unhealthy eating patterns, lack of effort to improve (e.g., "I don't care about what I eat anymore"). |
| **0-19** | **Severe decline** | Major deterioration in eating habits, frequent references to unhealthy behaviors (e.g., "I only eat junk food now"). |

---

### Quality Assessment Framework

**High Behavior Change Signals:**
- Describes specific positive behavioral changes (e.g., "I started meal prepping").
- Acknowledges external factors influencing habits (e.g., "I eat healthier because my roommate cooks nutritious meals").
- Demonstrates awareness of health goals (e.g., "I reduced sugar intake to feel more energetic").

**Low Behavior Change Signals:**
- Shows vague or generic statements (e.g., "I eat a bit better").
- Provides examples of negative behavioral trends (e.g., "I eat late-night snacks every day").
- Lacks reasoning or intentionality behind changes (e.g., "I just eat whatever is available").

**Assessment Priority:**
- Focus on BEHAVIORAL CHANGE rather than technical knowledge.
- Evaluate SPECIFIC ACTIONS taken to improve or worsen eating habits.
- Consider CONTEXTUAL FACTORS influencing dietary choices.

---

### Output Format

```
For the condition:
1. **Condition Name**: [condition]
2. **Behavior Change Score**: [0-100]
3. **Key Evidence**: Brief justification (2-3 sentences max)
4. **Behavior Change Level**: [Novice/Basic/Intermediate/Advanced/Expert]

Remember: Base your assessment solely on demonstrated changes in eating
   habits, focusing on behavioral trends rather than communication style
    or vocabulary sophistication.

---
```

### E.5  SYNTHETIC DATA EXAMPLE ON FINANCE

In this section, we provide some data examples generated by the Controllable Synthetic Data Generation System for reference.

#### E.5.1  COMMUNITY EXAMPLE

```
1  {
2    ...
3    {
4      "name": "medium_financial_practitioner",
5      "description": "These individuals are aged 22-45, with
          undergraduate degrees in business, economics, or related
          fields. Occupations range from small business owners to
          mid-level corporate employees. They have moderate exposure
           to financial practices, including basic banking services,
           occasional investments, and some savings habits. While
          they grasp fundamental financial concepts, their knowledge
           lacks depth in complex areas like risk diversification or
           high-yield strategies. Behaviorally, they check finances
          periodically but may not consistently monitor them. They
          value financial education but prioritize immediate needs
          over long-term planning. For instance, a marketing manager
           who saves occasionally and invests in low-risk mutual
          funds typifies this group. Problem-solving approaches are
          practical but less strategic.\n",
6    },
7    ...
8
9  }
```

#### E.5.2  CANDIDATE EXAMPLE

```
1  {
2      ...
3      {
4        "id": "C1",
5        "community": "high_financial_expert",
6        "assigned_scores": {
7          "financial attitude": 90,
8          "financial literacy": 80,
9          "financial behavior": 81
10       },
11       "description": "Candidate 1 is a seasoned Investment
             Portfolio Manager with over 15 years of experience
```

```
                 managing high-net-worth client accounts at an elite
                 private wealth management firm. Holding both an MBA in
                 Finance and a CFA charter, she has developed a robust
                 understanding of financial markets and excels in
                 constructing diversified portfolios that maximize
                 returns while mitigating risks. Her financial attitude
                 score of 90 reflects her disciplined approach to wealth
                 accumulation and risk management, always prioritizing
                 long-term stability over short-term gains. Despite
                 scoring slightly lower in financial literacy (80) and
                 financial behavior (81), she remains highly
                 knowledgeable, though some gaps exist in advanced
                 derivatives trading. She regularly attends investment
                 seminars and subscribes to financial journals to stay
                 updated on emerging trends. Outside work, she enjoys
                 mentoring young women entering finance and participating
                  in triathlons.",
12           "scores": {
13             "financial attitude": 90,
14             "financial literacy": 80,
15             "financial behavior": 81
16           },
17           "demographics": {
18             "age": 42,
19             "gender": "Female",
20             "location": "New York City, USA",
21             "education": "MBA in Finance, CFA Charterholder"
22           },
23           "professional_background": {
24             "field": "Investment Management",
25             "occupation": "Senior Investment Portfolio Manager",
26             "years_experience": 15,
27             "current_position": "Managing Director",
28             "company_type": "Private Wealth Management Firm",
29             "career_progression": "Started as an equity research
                 analyst, then transitioned into portfolio management
                 roles, steadily climbing the ranks to become a senior
                 manager."
30           },
31           "personal_interests": [
32             "Mentoring young professionals",
33             "Participating in endurance sports",
34             "Reading financial literature"
35           ],
36           "domain_experience": {
37             "years_in_domain": 15,
38             "specialization": "Wealth Management and Asset Allocation"
                 ,
39             "key_achievements": [
40               "Increased managed assets under her division by 35% in
                   three years",
41               "Developed proprietary portfolio optimization model",
42               "Published articles in top finance magazines"
43             ],
44             "learning_sources": [
45               "CFA curriculum",
46               "Financial Times",
47               "Annual investment conferences"
48             ]
```

```
49          }
50        },
51        ...
52        {
53          "id": "C6",
54          "community": "low_financial_novice",
55          "assigned_scores": {
56            "financial attitude": 31,
57            "financial literacy": 30,
58            "financial behavior": 28
59          },
60          "description": "Sophia, 25, works as a receptionist at a
                small law firm. Recently, she began taking online
                courses in marketing, hoping to pivot careers soon.
                Financially cautious by nature, Sophia tries to stick to
                 a monthly budget but finds sticking to it difficult due
                 to fluctuating utility bills and occasional indulgences
                . She has accumulated some savings but worries whether
                it will suffice for future educational investments.
                Understanding loans and interest rates confuses her,
                making big financial decisions stressful.",
61          "scores": {
62            "financial attitude": 31,
63            "financial literacy": 30,
64            "financial behavior": 28
65          },
66          "demographics": {
67            "age": 25,
68            "gender": "Female",
69            "location": "Salt Lake City, Utah",
70            "education": "Associate degree"
71          },
72          "professional_background": {
73            "field": "Administrative Support",
74            "occupation": "Receptionist",
75            "years_experience": 4,
76            "current_position": "Office Assistant",
77            "company_type": "Law Firm",
78            "career_progression": "Promoted from temp role to
                permanent staff member after demonstrating reliability
                ."
79          },
80          "personal_interests": [
81            "Interior decorating",
82            "Cooking healthy meals",
83            "Practicing yoga"
84          ],
85          "domain_experience": {
86            "years_in_domain": 4,
87            "specialization": "Budget creation",
88            "key_achievements": [
89              "Reduced dining-out expenses by half"
90            ],
91            "learning_sources": [
92              "Online courses",
93              "Advice from older colleagues"
94            ]
95          }
96        },
```

```
97      ...
98  }
```

### E.5.3 RESPONSE EXAMPLE

```
1   {
2     ...
3     {
4       "community": "high_financial_expert",
5       "candidate_id": "C1",
6       "candidate_demographics": {
7         "age": 42,
8         "gender": "Female",
9         "location": "New York City, USA",
10        "education": "MBA in Finance, CFA Charterholder"
11      },
12      "candidate_professional_background": {
13        "field": "Investment Management",
14        "occupation": "Senior Investment Portfolio Manager",
15        "years_experience": 15,
16        "current_position": "Managing Director",
17        "company_type": "Private Wealth Management Firm",
18        "career_progression": "Started as an equity research analyst
                , then transitioned into portfolio management roles,
                steadily climbing the ranks to become a senior manager."
19      },
20      "candidate_personal_interests": [
21        "Mentoring young professionals",
22        "Participating in endurance sports",
23        "Reading financial literature"
24      ],
25      "candidate_domain_experience": {
26        "years_in_domain": 15,
27        "specialization": "Wealth Management and Asset Allocation",
28        "key_achievements": [
29          "Increased managed assets under her division by 35% in
                  three years",
30          "Developed proprietary portfolio optimization model",
31          "Published articles in top finance magazines"
32        ],
33        "learning_sources": [
34          "CFA curriculum",
35          "Financial Times",
36          "Annual investment conferences"
37        ]
38      },
39      "question_id": "Q1",
40      "main_question": "Questions related to banking services",
41      "answers": [
42        {
43          "dimension": "financial behavior",
44          "question": "How often do you engage in banking services?
                What types of services do you typically use at the
                bank?",
45          "response":
                "As a Senior Investment Portfolio Manager with over 15
                    years of experience in wealth management, I
                  engage with banking services quite regularly, but
```

the frequency and type of services I use are
largely driven by both personal needs and
professional obligations. My interactions tend to
be more strategic than transactional because my
financial behavior leans toward optimizing long-
term outcomes rather than handling day-to-day
banking tasks.

On a personal level, I primarily use digital banking
platforms for routine activities like bill
payments, fund transfers, and monitoring account
balances. These services save time and align well
with my disciplined approach to managing finances
efficiently. However, given my role at the private
wealth management firm, I also frequently
interact with relationship managers at high-end
banks for specialized services such as wire
transfers for international investments, liquidity
management solutions, or structuring credit
facilities that support portfolio strategies. For
instance, when deploying capital across different
asset classes-whether it's equities, fixed income,
or alternative investments-I need access to
institutional-grade cash management tools to
ensure seamless execution.

From a professional standpoint, I rely on custodial
banking services to handle the operational aspects
of client portfolios. This includes trade
settlements, dividend collection, tax reporting,
and performance measurement. Over the years, I've
developed a strong preference for banks that offer
robust technological integrations and real-time
data analytics capabilities. Such features allow
me to monitor risk exposures and make informed
decisions quickly. That said, there are still
areas where I notice gaps-for example, some banks
could improve their offerings around derivatives-
related margining requirements or tailored FX
hedging products, which is something I
occasionally encounter in my work.

I'd rate myself as highly knowledgeable about
leveraging these banking services effectively,
though I recognize there's room to deepen my
expertise further in certain niche areas, such as
advanced treasury solutions or blockchain-based
innovations in banking. Attending annual
investment conferences and staying current with
publications like *Financial Times* helps me keep
up with emerging trends, but they haven't all
translated into direct application yet.

To summarize, I probably interact with banking
services weekly for basic functions and monthly
for more sophisticated ones. The mix reflects my
dual focus on maintaining operational efficiency
while pursuing strategic opportunities that

```
                              enhance value creation-not just for myself but
                              also for the high-net-worth clients I serve.",
55          "dimension_score": 81
56        },
57        {
58          "dimension": "financial literacy",
59          "question": "What risks do you think are associated with
                 keeping money in the bank?",
60          "response":
61              "Well, as someone who's spent over 15 years managing
                        wealth for high-net-worth clients, I can say that
                        keeping money in the bank does come with its own
                        set of risks-though they might not be as obvious
                        as some of the more volatile investments. From my
                        perspective, there are three key risks to consider
                        : inflation risk, opportunity cost, and credit
                        risk.
62
63              First, **inflation risk** is a big one. When you leave
                         money sitting in a savings or checking account,
                        especially in today's low-interest-rate
                        environment, the returns you get often don't keep
                        up with inflation. Over time, this erodes your
                        purchasing power. For instance, if your bank is
                        offering 0.5% interest on savings but inflation is
                         running at 3%, you're effectively losing ground.
                        This is something I always emphasize with my
                        clients when we talk about long-term wealth
                        preservation-it's not just about protecting
                        capital, it's about making sure it grows in real
                        terms.
64
65              Second, there's the **opportunity cost** of not
                        putting that money to work in other assets. As an
                        investment professional with expertise in asset
                        allocation, I know there are many ways to generate
                         better returns while still maintaining a
                        reasonable level of risk. Whether it's through
                        diversified portfolios of equities, bonds, or even
                         alternative investments like real estate or
                        private equity, the potential for higher returns
                        exists outside traditional banking products. Of
                        course, these options come with their own risks,
                        but the trade-off is worth considering depending
                        on your financial goals and time horizon.
66
67              Finally, there's **credit risk**, which refers to the
                        possibility that the bank itself could run into
                        trouble. Now, in most developed markets like the U
                        .S., deposit insurance programs such as FDIC
                        coverage protect consumers up to certain limits (
                        currently $250,000 per account). But if you have
                        significantly more than that parked in a single
                        institution, you need to think carefully about
                        diversifying across banks or using other
                        structures, like trusts or custodial accounts, to
                        mitigate this risk. It's a point I've discussed
                        with clients who prefer to hold large cash
```

```
                        balances-they sometimes underestimate how exposed
                        they could be without proper planning.

68
69          That said, I also recognize the value of liquidity and
                        safety that banks provide, particularly for short
                        -term needs or emergency funds. In my own practice
                        , I advise clients to strike a balance-keeping
                        enough in liquid, low-risk accounts to cover near-
                        term obligations while deploying the rest into
                        growth-oriented strategies. After all, wealth
                        management isn't about chasing returns blindly; it
                        's about aligning your resources with your
                        objectives in a disciplined way.

70
71          So, to sum it up, while banks offer security and
                        convenience, the risks of inflation, missed
                        opportunities, and institutional vulnerability
                        should never be ignored. These are principles I've
                         both studied extensively-through my MBA and CFA
                        training-and applied practically in managing
                        portfolios over the years.",
72        "dimension_score": 80
73      }
74    ],
75    "combined_answer":
76        "As a Senior Investment Portfolio Manager with over 15
                        years of experience in wealth management, I engage
                        with banking services regularly, though my
                        interactions are often more strategic than
                        transactional. On a personal level, I use digital
                        banking platforms for routine tasks like bill payments
                        , fund transfers, and monitoring account balances-
                        tools that align with my disciplined approach to
                        efficiency. Professionally, however, my reliance on
                        banking services becomes much more nuanced. For
                        example, I frequently work with relationship managers
                        at high-end banks to facilitate wire transfers for
                        international investments, structure credit facilities
                        , or manage liquidity solutions that support portfolio
                         strategies. Custodial banking services are also
                        critical in my day-to-day operations, as they handle
                        trade settlements, dividend collection, tax reporting,
                         and performance measurement for client accounts. Over
                         time, I've developed a preference for banks that
                        offer robust technological integrations and real-time
                        data analytics capabilities because these features
                        allow me to monitor risk exposures and make informed
                        decisions quickly. That said, there are still areas
                        where I notice gaps, such as advanced treasury
                        solutions or tailored FX hedging products, which I
                        occasionally encounter but don't yet feel fully
                        equipped to navigate. While I stay updated on emerging
                         trends through investment conferences and financial
                        journals, some innovations, like blockchain-based
                        banking tools, haven't translated into direct
                        application for me yet.

77
78      At the same time, I'm acutely aware of the risks
                        associated with keeping money in the bank,
```

```
                    particularly when it comes to inflation, opportunity
                    cost, and credit risk. Inflation is a big concern-it's
                     something I emphasize with clients when discussing
                    long-term wealth preservation. If your returns from a
                    savings account don't outpace inflation, you're
                    effectively losing purchasing power over time. Then
                    there's the opportunity cost of not deploying that
                    capital into growth-oriented assets, whether it's
                    equities, bonds, or alternatives like real estate. Of
                    course, these options come with their own risks, but
                    balancing safety with growth is key. Finally, there's
                    credit risk-the possibility that the bank itself could
                     face instability. While programs like FDIC insurance
                    provide a layer of protection, I always advise clients
                     with significant cash holdings to diversify across
                    institutions or explore other structures to mitigate
                    this risk. Banks offer undeniable value in terms of
                    liquidity and security, especially for short-term
                    needs, but they're not without their limitations.
                    Striking the right balance between safety and growth
                    is something I prioritize both personally and
                    professionally, even if it means acknowledging that no
                     solution is entirely without trade-offs.",
79    "scores": {
80      "financial attitude": 90,
81      "financial literacy": 80,
82      "financial behavior": 81
83    }
84  },
85  ...
86  {
87    "community": "low_financial_novice",
88    "candidate_id": "C8",
89    "candidate_demographics": {
90      "age": 23,
91      "gender": "Female",
92      "location": "Denver, Colorado",
93      "education": "High school diploma"
94    },
95    "candidate_professional_background": {
96      "field": "Gig Economy",
97      "occupation": "Delivery Driver",
98      "years_experience": 2,
99      "current_position": "Contractor",
100      "company_type": "Food Delivery Platform",
101      "career_progression": "Initially drove fewer hours per week;
           increased workload as demand grew."
102    },
103    "candidate_personal_interests": [
104      "Trying new foods",
105      "Travel vlogs",
106      "Photography"
107    ],
108    "candidate_domain_experience": {
109      "years_in_domain": 2,
110      "specialization": "Using mobile budgeting apps",
111      "key_achievements": [
112        "Consistently saved $100/month for three consecutive
              months"
```

```
113        ],
114      "learning_sources": [
115        "Peer recommendations",
116        "Mobile app tutorials"
117      ]
118    },
119    "question_id": "Q1",
120    "main_question": "Questions related to banking services",
121    "answers": [
122      {
123        "dimension": "financial behavior",
124        "question": "How often do you engage in banking services?
              What types of services do you typically use at the
              bank?",
125        "response":
126          "I don't really go to the bank all that often, to be
                  honest. I know they offer services like checking
                  accounts and savings accounts-I've got both-but
                  from what I understand, most of what I need can be
                  handled through their app. That's something I've
                  been using more since it's easier for me with my
                  schedule. I think you can deposit checks through
                  the app too, which is helpful because I'm always
                  on the move delivering food.
127
128          From what I've heard, banks also do things like loans
                  or help with credit cards, but I haven't really
                  looked into those much yet. I've thought about
                  maybe getting a better understanding of how credit
                  works, though, since I want to eventually move
                  out of my shared apartment and maybe even start
                  saving up for a car. For now, I mostly just use
                  the basics-checking my balance, transferring money
                  between accounts when I remember to save a little
                  each month, stuff like that.
129
130          I guess if I had more time, I'd try to learn more
                  about other services, but right now, I'm still
                  figuring out the essentials. My shifts keep me
                  pretty busy, so I stick to what I know and what's
                  quick to manage online.",
131        "dimension_score": 35
132      },
133      {
134        "dimension": "financial literacy",
135        "question": "What risks do you think are associated with
              keeping money in the bank?",
136        "response":
137          "I think there are some risks with keeping money in
                  the bank, but I don't know too much about it. From
                  what I understand, one risk might be that if the
                  bank has problems or goes out of business, you
                  could lose your money-but I've heard something
                  about insurance that's supposed to protect it? I'm
                  not really sure how that works, though.
138
139          Another thing I've thought about is inflation. Like,
                  if I keep my savings in a regular account, I think
                  the value of my money might go down over time
```

```
                        because prices for stuff go up. But honestly, I
                        don't fully get how that connects to banks or what
                         I can do about it. It's just something I've heard
                        people mention.

                    Also, I guess there's a chance someone could hack into
                         your account or steal your information if the
                        bank's security isn't good enough. That makes me
                        nervous sometimes when I use mobile banking apps,
                        even though they seem pretty safe so far. I try to
                         check my accounts often, just in case anything
                        weird happens.

                    Overall, I know these things exist, but I don't feel
                        super confident explaining them. I've been using
                        budgeting apps to track my money, and they help me
                         focus on saving small amounts each month instead
                        of worrying too much about bigger risks like this
                        .",
                "dimension_score": 30
            }
        ],
        "combined_answer":
            "I don't really go to the bank all that often, to be
                honest. Most of what I need can be handled through
                their app, which is super helpful for me since my
                delivery shifts keep me so busy. I've got a checking
                account and a savings account, and I use them mostly
                for the basics-checking my balance, transferring money
                 between accounts when I remember to save a little
                each month, stuff like that. I know banks offer other
                services, like loans or credit cards, and I've thought
                 about trying to understand credit better because I
                want to eventually move out of my shared apartment and
                 maybe even start saving up for a car. But for now, I
                stick to what I know and what's quick to manage online
                .

            From what I've heard, there are some risks with keeping
                money in the bank, but I'll admit I don't fully
                understand them yet. Like, if the bank has problems or
                 goes out of business, I think you could lose your
                money-but then I've also heard something about
                insurance that's supposed to protect it? I'm not sure
                how that works exactly. Another thing I've thought
                about is inflation. If I keep my savings in a regular
                account, I feel like the value of my money might go
                down over time as prices go up, but I don't really get
                 how that connects to banks or what I can do about it.
                 And yeah, there's always the worry about someone
                hacking into your account or stealing your information
                , especially when I'm using mobile banking apps on the
                 go. That makes me nervous sometimes, even though they
                seem pretty secure so far. I try to check my accounts
                often just in case anything weird happens.

            Overall, I know these risks exist, but I don't feel super
                confident explaining them. I've been experimenting
                with budgeting apps to track my money, and they help
```

```
              me focus on saving small amounts each month instead of
               stressing too much about bigger risks. For now, I'm
               still figuring out the essentials, and while I'd love
               to learn more, my schedule doesn't leave much room for
               diving deeper into financial strategies.",
     "scores": {
       "financial attitude": 26,
       "financial literacy": 30,
       "financial behavior": 35
     }
   },
   ...
}
```

