# OpenReview forum: "IntE: Quantitative Framework for Qualitative Data Evaluation via Distributional Mining"
_ICLR.cc/2026/Conference — ICLR 2026 Conference Withdrawn Submission_

### Official Review · Reviewer_8gwX · 2025-10-29

**Soundness:** 3
**Presentation:** 3
**Contribution:** 2
**Rating:** 4
**Confidence:** 3

**Summary:**

The paper introduces IntE, a quantitative framework for evaluating qualitative response datasets. It compares demographic distributions with LLM-derived intrinsic clusters to measure how well a dataset captures general patterns versus unique insights. The framework defines four metrics—GMR, DDR, DP, and DC—and includes an anchor-based LLM evaluation system and instruction optimization process. Experiments on synthetic and real datasets show that these metrics can characterize dataset diversity and representativeness.

**Strengths:**

1. The idea of evaluating “dataset potential” via alignment between intrinsic and extrinsic distributions is genuinely new. It formalizes what qualitative researchers usually discuss intuitively.

2. Clear decomposition. The framework is modular and easy to follow — instruction optimization, dissimilarity computation, clustering, and four metrics. Figures and examples make the workflow understandable.

3. Interdisciplinary relevance. Bridges qualitative data analysis, data valuation, and LLM evaluation — with good potential for social science or HCI applications.

**Weaknesses:**

1. Limited benchmarking. There’s no quantitative comparison to standard dataset or clustering metrics (e.g., Silhouette, ARI/V-measure, dataset cartography). Without such baselines, it’s hard to judge how much IntE truly adds beyond reinterpreting known measures.


2. Approximation validity. For large datasets, the paper replaces pairwise distances with scalar differences but doesn’t show how much information is lost. There’s no theoretical bound or empirical error curve.

3. Real-world evidence too thin. The single food-choice case study feels anecdotal; there’s no cross-domain validation or expert consistency check.

**Questions:**

1. Could you provide quantitative comparisons with existing data quality or clustering validity metrics to show the unique advantage of IntE?

2. How consistent are the LLM-based distance measures across random seeds, temperatures, or different LLMs? Please provide variance or confidence intervals.

3. In large-scale approximation, how large is the deviation from true pairwise distance? Any error analysis or theoretical argument?

---

### Official Review · Reviewer_uJ4p · 2025-10-31

**Soundness:** 2
**Presentation:** 2
**Contribution:** 1
**Rating:** 2
**Confidence:** 3

**Summary:**

This paper introduce a framework for analysis of qualitative response datasets. The framework combine cluster analysis and multi-agent LLMs to evaluate the datasets.

**Strengths:**

Paper is generally easy to follow and have experimented with different LLMs on both synthetic and real-world datasets. Both computational and human evaluation are conducted.

**Weaknesses:**

1. Lack of discussion on related works. a) Papers on computational qualitative analysis such as content analysis or thematic analysis with LLM. b) Recent work which leverage multi-agent system to analyse qualitative datasets. c) author only mention the limitation of LLM as judge method without discussing which type of llm as judge this paper is similar to.
2. The paper claim to achieve quantitative evaluation of qualitative dataset, however, the framework appear to be qualitative evaluation with LLM which is enhanced by cluster analysis.
3. There is lack of detail on which clustering methods are used and which encoders are used, the hyperparameters of clustering are not discussed or experimented. This is concerning as the cluster quality depend on these. The cluster quality is not well evaluated either.
4. There is lack of justification for the design of each components within the framework. Why are each component nessecery? It is also unclear why the assessment need to be four quadrant.
5. There is lack of comparison to other published multi-agent for analysing qualitative datasets in the experiment section. The novel contribution compare to existing work is unclear.
6. Lack of detail on human evaluation and whether their evaluation method is supported by social science.

**Questions:**

See weakness. I will increase the score accordingly if the authors address most or part of the concerns.

---

### Official Review · Reviewer_nYgZ · 2025-11-01

**Soundness:** 2
**Presentation:** 2
**Contribution:** 2
**Rating:** 2
**Confidence:** 4

**Summary:**

In this paper, the authors introduce IntE, a framework for quantitatively assessing qualitative response datasets. IntE uses a four-quadrant assessment procedure that quantifies the utility of a dataset containing semi-structured human responses. IntE is evaluated with controlled experiments on synthetic data as well as a real-world social survey use case.

**Strengths:**

- This paper addresses an important and under-researched challenge: assessing the quality of human response datasets in a principled manner
- The authors’ focus on deriving both general patterns and unique insights (encompassing both the distribution-level and the single sample-level) is interesting and likely to be useful

**Weaknesses:**

- **Utility of proposed approach:**
    - The authors cluster data and then align clusters with predefined demographic groups; however, the motivation behind why this approach yields insights into the *quality* of the dataset is not clear
    - The proposed approach has lots of components and many hyperparameters, which are likely to limit usability in the real-world
- **Weak evaluations and insufficient analysis:** The proposed approach is not thoroughly evaluated nor compared to baselines
    - For the synthetic analysis, no information on the synthetic framework is provided in the main text, making it challenging to understand the results. Moreover, the majority of the results here (Sections 4.1.1 and 4.1.2) primarily report results from ablations. The key results on overall performance are relegated to a small section on Page 9 (Section 4.1.3) and are not analyzed in detail. Moreover, it is not clear to me how the results in 4.1.3 align with the authors’ central claim in this paper that their proposed method evaluates dataset *quality*.
    - Similarly, for the real-world case study, no analysis or insights into the dataset are provided. The authors claim that the “resulting metrics…faithfully represented the data’s quality.” However, this claim is not supported, and it is unclear to me how the provided metrics provide insight into data quality. This real-world case study does not effectively demonstrate how IntE can accelerate knowledge discovery in practice.
- **Writing quality:** The structure of this paper, specifically in Sections 3 and 4, is confusing, making it challenging to understand the key contributions of this work

**Questions:**

My questions are listed above in the weaknesses section

---

### Official Review · Reviewer_9YJh · 2025-11-02

**Soundness:** 2
**Presentation:** 2
**Contribution:** 1
**Rating:** 2
**Confidence:** 3

**Summary:**

This paper presents a framework to evaluate the quality of questionnaire answers. The assumption is that answers should be somewhat similar to what is expected based on the interviewee’s demographic information. The framework highlights two technical approaches. The first is an LLM-based dissimilarity calculation, guided by the LLM-as-a-judge paradigm, and incorporates an iterative query refinement algorithm. The second involves handcrafted dataset quality assessment metrics, based on the assumption that the sample distribution forms a clear clustering structure.

**Strengths:**

- The problem setting is novel and interesting. The proposed approach could potentially impact broader non-ML fields such as sociology and policy design.

- The proposed query refinement approach has the potential to address the arbitrariness of LLM queries in this problem setting.

**Weaknesses:**

As an ML paper, the only technical highlight is the iterative query refinement approach. However, the description is vague, and given the diversity of possible queries, it is questionable whether such an approach truly works. My concern is amplified by the fact that the authors didn't mention any specific initialization method for the iterative algorithm.

More importantly, the authors do not clearly validate their key assumptions, such as: (1) questionnaire answers result in clear clustering structures, and (2) the distribution of questions is predictable from demographic information to enable the alignment-based evaluation.

Overall, the framework appears to be based primarily on the authors’ speculations rather than on evidence established through scientific methods. As an ML paper, I conclude that it makes limited contributions and does not meet the acceptance threshold.

**Questions:**

- How do you initialize the iterative algorithm in (2).

- How do you determine $k_\text{max}$? Is it given?

- How do you define the sorted anchor manifold?

---

### Note · Authors · 2025-12-22

I have read and agree with the venue's withdrawal policy on behalf of myself and my co-authors.